# Competition in the chaperone-client network subordinates cell-cycle entry to growth and stress

David F Moreno[1,*], Eva Parisi[1,*], Galal Yahya[1,2,*], Federico Vaggi[3], Attila Csikász-Nagy[4,5], Martí Aldea[1,6]

**The precise coordination of growth and proliferation has a universal prevalence in cell homeostasis. As a prominent property, cell size is modulated by the coordination between these processes in bacterial, yeast, and mammalian cells, but the underlying molecular mechanisms are largely unknown. Here, we show that multifunctional chaperone systems play a concerted and limiting role in cell-cycle entry, specifically driving nuclear accumulation of the G1 Cdk–cyclin complex. Based on these findings, we establish and test a molecular competition model that recapitulates cell-cycle-entry dependence on growth rate. As key predictions at a single-cell level, we show that availability of the Ydj1 chaperone and nuclear accumulation of the G1 cyclin Cln3 are inversely dependent on growth rate and readily respond to changes in protein synthesis and stress conditions that alter protein folding requirements. Thus, chaperone workload would subordinate Start to the biosynthetic machinery and dynamically adjust proliferation to the growth potential of the cell.**

## Introduction

Under unperturbed conditions, growth cells maintain their size within constant limits, and different pathways have concerted roles in processes leading to growth and proliferation (Cook & Tyers, 2007; Marshall et al, 2012; Turner et al, 2012). Here, we will use the term growth to refer to cell mass or volume increase, whereas the term proliferation will be restricted to the increase in cell number. Cell growth is dictated by many environmental factors in budding yeast, and the rate at which cells grow has profound effects on their size. High rates of macromolecular synthesis promote growth and increase cell size. Conversely, conditions that reduce cell growth limit macromolecular synthesis and reduce cell size. This behavior is nearly universal, and it has been well characterized in bacteria, yeast, diatoms, and mammalian cells of different origins (Aldea et al, 2017). A current view sustains that cell cycle and cell growth machineries should be deeply interconnected to ensure cell homeostasis and adaptation, but the causal molecular mechanism is still poorly understood (Lloyd, 2013).

In budding yeast, cyclin Cln3 is the most upstream activator of Start (Tyers et al, 1993). Cln3 forms a complex with Cdc28, the cell-cycle Cdk in budding yeast, and activates the G1/S regulon with the participation of two other G1 cyclins, Cln1 and Cln2, which contribute to phosphorylate the Whi5 inhibitor, thus creating a positive feedback loop that provides Start with robustness and irreversibility (Bertoli et al, 2013). The Start network in mammals offers important differences, particularly in the structure and number of transcription factors, but the core of the module is strikingly similar, where Cdk4,6–cyclin D complexes phosphorylate RB and activate E2F-DP transcription factors in a positive feedback loop involving Cdk2–cyclin E (Bertoli et al, 2013).

As they are intrinsically unstable, G1 cyclins are thought to transmit growth information for adapting cell size to environmental conditions. The Cln3 cyclin is a dose-dependent activator of Start (Sudbery et al, 1980; Nash et al, 1988; Cross & Blake, 1993) that accumulates in the nucleus because of a constitutive C-terminal NLS (Edgington & Futcher, 2001; Miller & Cross, 2001) and the participation of Hsp70-Hsp40 chaperones, namely Ssa1,2 and Ydj1 (Vergés et al, 2007). In addition, Ssa1 and Ydj1 also regulate Cln3 stability (Yaglom et al, 1996; Truman et al, 2012) and play an essential role in setting the critical size as a function of growth rate (Ferrezuelo et al, 2012). In mammalian cells, cyclin D1 depends on Hsp70 chaperone activity to form trimeric complexes with Cdk4 and NLS-containing KIP proteins (p21, p27, and p57) that drive their nuclear accumulation (Diehl et al, 2003).

Molecular chaperones assist nascent proteins in acquiring their native conformation and prevent their aggregation by constraining non-productive interactions. These specialized folding factors also guide protein transport across membranes and

[1]Molecular Biology Institute of Barcelona, CSIC, Catalonia, Spain  [2]Department of Microbiology and Immunology, Zagazig University, Zagazig, Egypt  [3]Department of Informatics, Ecole Normale Supérieure, INRIA, Sierra Team, Paris, France  [4]Randall Centre for Cell and Molecular Biophysics and Institute of Mathematical and Molecular Biomedicine, King's College London, London, UK  [5]Pázmány Péter Catholic University, Faculty of Information Technology and Bionics, Budapest, Hungary  [6]Department of Basic Sciences, Universitat Internacional de Catalunya, Sant Cugat del Vallès, Spain

Correspondence: attila.csikasz-nagy@kcl.ac.uk; marti.aldea@ibmb.csic.es
Federico Vaggi's present address is Amazon, Seattle, WA 98121, USA
*David F Moreno, Eva Parisi, and Galal Yahya contributed equally to this work

modulate protein complex formation by controlling conformational changes (Kampinga & Craig, 2010). Chaperones are involved in key growth-related cellular processes, such as protein folding and membrane translocation during secretion (Kim et al, 2013), and many chaperone-client proteins have crucial roles in the control of growth, cell division, environmental adaptation, and development (Gong et al, 2009; Taipale et al, 2012, 2014). Thus, because chaperones required for Cdk–cyclin activation are also involved in the vast majority of processes underlying cell growth, we hypothesized that competition for shared multifunctional chaperones could subordinate entry into the cell cycle to the biosynthetic machinery of the cell.

Here, we show that chaperones play a concerted and limiting role in cell-cycle entry, specifically driving nuclear accumulation of the G1 Cdk–cyclin complex. Ydj1 availability is inversely dependent on growth rate and, based on our findings, we have established a molecular competition model that recapitulates cell-cycle-entry dependence on growth rate. As key predictions of the model, we show that nuclear accumulation of the G1 cyclin Cln3 is negatively affected by growth rate in a chaperone-dependent manner and rapidly responds to conditions that perturb or boost chaperone activity. Thus, chaperone availability would act as a G1 Cdk modulator transmitting both intrinsic and extrinsic information to subordinate Start and the critical size to the growth potential of the cell.

## Results

### Nuclear accumulation of the G1 Cdk depends on chaperone activity

Cln3 contains a bipartite nuclear localization signal at its C terminus that is essential for timely entry into the cell cycle (Edgington & Futcher, 2001; Miller & Cross, 2001), and we had found that the Ydj1 chaperone is important for nuclear accumulation of Cln3 (Vergés et al, 2007) and for setting the critical size as a function of growth rate (Ferrezuelo et al, 2012). Thus, we decided to characterize the role of chaperones as regulators of the G1 Cdk during G1 progression at a single-cell level in time-lapse experiments. Cln3 is too short-lived to be detected as a fluorescent protein fusion in single cells unless stabilized mutants are used (Liu et al, 2015; Schmoller et al, 2015). As previously described, mCitrine-Cln3-11A displayed a distinct nuclear signal in most asynchronously growing cells, arguing against the ER retention mechanism that we had proposed previously (Vergés et al, 2007). However, this protein is much more stable compared with wild-type Cln3, and transient retention at the ER would likely be obscured by accumulation of abnormally stable mCitrine-Cln3-11A in the nucleus. Thus, we decided to test whether nuclear accumulation of this stabilized protein was still dependent on Ydj1 by carefully measuring nuclear and cytoplasmic fluorescence levels (Fig S1). Although overall levels as determined by immunoblotting were not altered, nuclear mCitrine-Cln3-11A levels strongly decreased in Ydj1-deficient cells (Fig 1A–C), which confirmed previous observations obtained with 3HA-tagged wild-type Cln3 by immunofluorescence (Vergés et al, 2007). Moreover,

whereas nuclear levels of Cdc28-GFP in late G1 cells were also negatively affected by deletion of *YDJ1*, overexpression of both Ydj1 and Ssa1 significantly increased the nuclear to cytoplasmic ratio of Cdc28-GFP both in mid- and late-G1 cells (Fig 1D). Likely because of the fact that Cdc28 is present at much higher levels than Cln3 (Tyers et al, 1993; Cross et al, 2002), differences in the steady-state nuclear levels of Cdc28 when comparing wild-type and Ydj1-deficient or overexpressing cells were only modest (Fig 1D). On the other hand, we cannot discard the effects of Cln1,2 cyclins synthesized at Start by the transcriptional feedback loop. Thus, we decided to analyze directly the import kinetics of Cdc28-GFP in G1 cells by nuclear fluorescence loss in photobleaching (FLIP) (Figs 1E, F, and S2). We found that, although being extremely dependent on Cln3 (Fig 1G), the nuclear import rate of Cdc28-GFP decreased in Ydj1-deficient cells and was clearly impaired when the chaperone function was compromised by azetidine 2-carboxylic acid (AZC), a proline analog that interferes with proper protein folding (Trotter et al, 2001) and causes large aggregates of misfolded proteins that sequester Ssa1 and other chaperones (Escusa-Toret et al, 2013). To test indirect effects through the constitutive nuclear import machinery, we analyzed a 2NLS-GFP construct fused to the estradiol-binding domain immediately after the addition of estradiol to allow nuclear import. As shown in Fig 1C and H, this construct did not show significant differences in nuclear localization or import kinetics in cells lacking Ydj1, Cln3, or in the presence of AZC. In all, our data are consistent with a chaperone-dependent mechanism that drives nuclear import of the Cdc28–Cln3 complex in G1 for the timely execution of Start.

### Multifunctional chaperones have a limiting role in setting cell size at budding

If chaperones have a role in coordinating cell growth and Start machineries, chaperone availability ought to be limiting for cell-cycle entry. Thus, we decided to test this proposition by increasing the gene dosage of different chaperone sets in low-copy centromeric vectors. We first analyzed the effects of Ssa1 and Ydj1 and observed that the budding volume of newborn cells with two copies of Ssa1 or Ssa1/Ydj1 was 5% and 11% smaller, respectively, compared with those with empty vector. In addition to the Hsp70 system (Ssa1 and Ydj1), we analyzed the effects of key components of the Hsp90 system (Hsc82 and Cdc37), which is important for holding the Cdk in a productive conformation for binding cyclins (Vaughan et al, 2006), and the segregase Cdc48 complex (Cdc48, Ufd1, and Npl4), which prevents degradation of ubiquitinated Cln3, and concurrently stimulates its ER release and nuclear accumulation to trigger Start (Parisi et al, 2018). Gene duplication in centromeric vectors increased chaperone gene expression by 1.5- to 2-fold in a specific manner (Fig S3A). Notably, although expression levels increased only modestly, additional copies of chaperone genes caused an additive reduction of the budding volume in asynchronous cultures (Fig 2A) and newborn daughter cells (Fig 2B). This effect was barely observed in large cells deficient for Cln3 (Fig 2B), suggesting that increased chaperone levels would reduce budding size by Cln3-dependent mechanisms. Whi5-deficient cells exhibited a small decrease in budding size in the presence of plasmids expressing the three chaperone sets, but less pronounced than that shown by

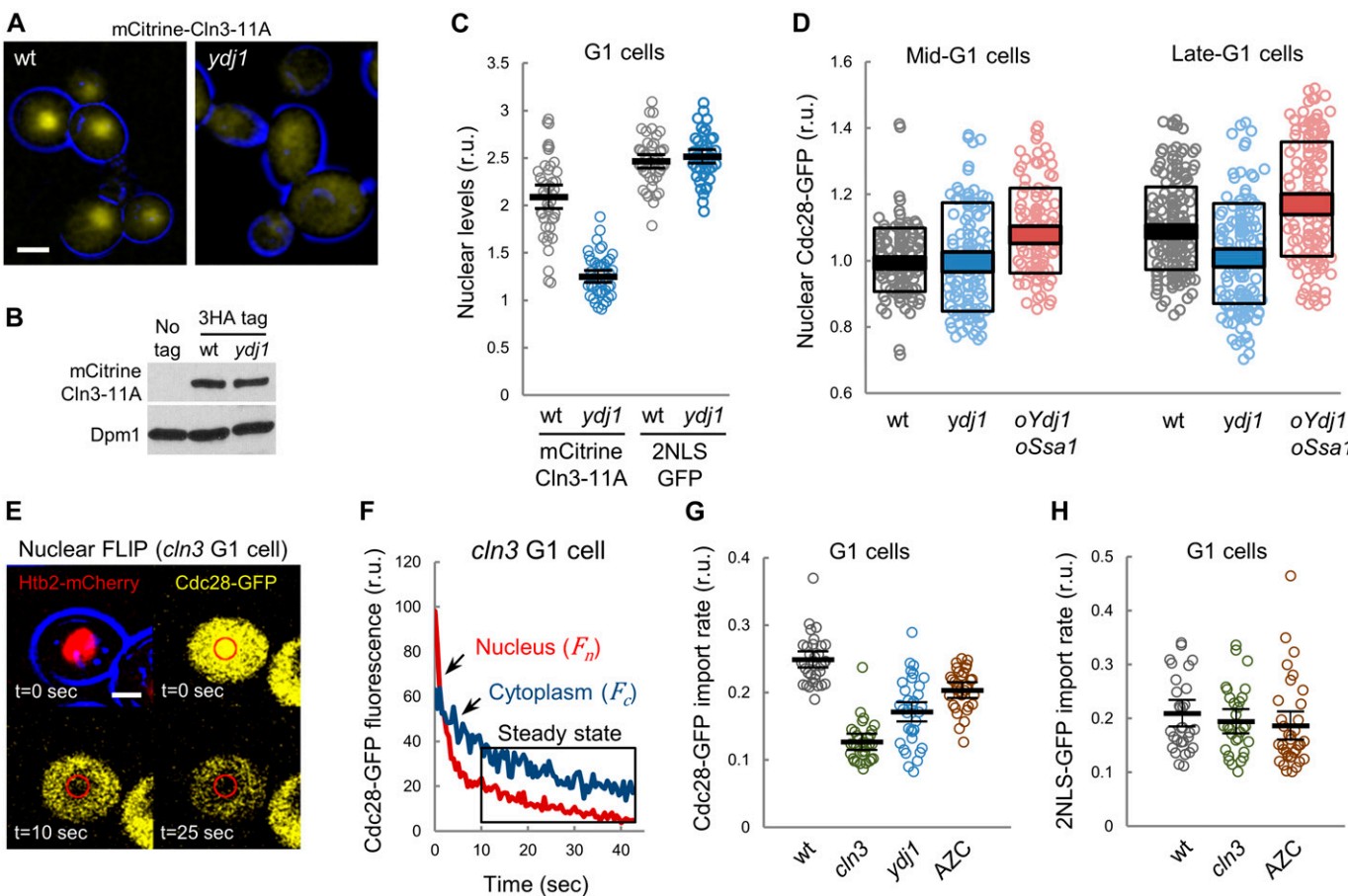

**Figure 1. Nuclear accumulation of the G1 Cdk depends on chaperone function.**
**(A)** Images corresponding to wt and Ydj1-deficient (*ydj1*) cells expressing mCitrine-Cln3-11A. Bar is 2 *μm*. **(B)** Immunoblotting analysis of 3HA-tagged mCitrine-Cln3-11A levels in wild-type and *ydj1* cells. Dpm1 is shown as loading control. **(C)** Nuclear to cytoplasmic mCitrine-Cln3-11A and 2NLS-GFP ratios for individual wild-type and *ydj1* G1 cells. Mean (N = 30, thick lines) and confidence limits ($α$ = 0.05, thin lines) for the mean are also plotted. **(D)** Cdc28-GFP wild-type (wt), Ydj1-deficient (*ydj1*), or overexpressing Ydj1 and Ssa1 (*oYdj1 oSsa1*) from the dual *GAL1-10p* promoter cells were analyzed by time-lapse microscopy during G1 progression. Mean (N > 100) nuclear to cytoplasmic Cdc28-GFP ratios are plotted with respective SD (white boxes) and confidence ($α$ = 0.05, colored boxes) intervals for the mean at either mid (36–45 min before start) or late G1 (18–21 min before start). **(E)** Analysis of Cdc-28-GFP import by nuclear FLIP. A representative *cln3* cell expressing Cdc28-GFP and Htb2-mCherry at different times during nuclear photobleaching is shown. Bar is 1 *μm*. **(F)** A representative nuclear FLIP output of Cdc28-GFP in a Cln3-deficient G1 cell showing fluorescence decay in nuclear and cytoplasmic compartments. **(G)** Cdc28-GFP import rates in wt, Cln3-deficient (*cln3*), and Ydj1-deficient (*ydj1*) single cells in the G1 phase. Wild-type cells treated with AZC are also shown. Mean values (N > 30, thick lines) with confidence limits for the mean ($α$ = 0.05, thin lines) are also plotted. **(H)** 2NLS-GFP-EBD import rates in wt and Cln3-deficient (*cln3*) G1 single cells after 5 min in the presence of 1 *μM* estradiol. Wild-type cells treated with AZC are also shown. Mean values (N > 30, thick lines) with confidence limits for the mean ($α$ = 0.05, thin lines) are also plotted.
Source data are available for this figure.

wt cells. Because *whi5* cells still require Cln3 to attain their small size (Jorgensen et al, 2002), the observed residual decrease could also be due to Cln3-mediated effects. Nonetheless, budding size reduction by increased chaperone expression was totally abolished in cells lacking Cln3 as well as Whi5 and Stb1 (Fig 2B), two key inhibitors of the Start network. This triple mutant displays un-altered average budding volume compared with wild type (Wang et al, 2009), but has lost most of the dependency on growth rate (Ferrezuelo et al, 2012). In contrast, the effect was still clear in cells lacking Whi7, an inhibitor of Start that acts at an upstream step (Yahya et al, 2014). Finally, we observed a comparable reduction in the budding volume when genes of the three chaperone systems were duplicated together in an artificial chromosome (Fig 2C). Doubling times of cells with vectors expressing the different

chaperone sets were not significantly different from wild-type cells. Also, we had found no differences in the growth rate during G1 when comparing wild-type and Ydj1/Ssa1 overexpressing cells (Ferrezuelo et al, 2012), which rules out indirect effects on cell size by growth rate changes. In addition, protein levels and phos-phorylation status of Cln3 were not affected by chaperone gene dosage (Fig S3B), which pointed to effects on Cdk–cyclin complex activity and/or localization. To give further support to these gene overexpression experiments, we carefully analyzed the effects of deleting one *YDJ1* copy in diploid cells on budding size. As shown in Fig S3, a reduction to ~60% in Ydj1 protein levels caused a significant and reproducible increase in budding size, thus supporting the limiting role of chaperones in setting cell size during cell-cycle entry.

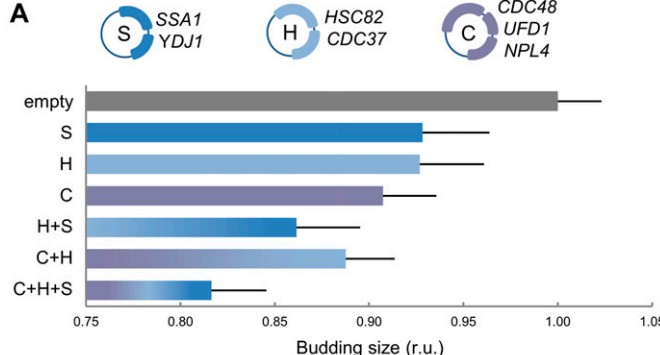

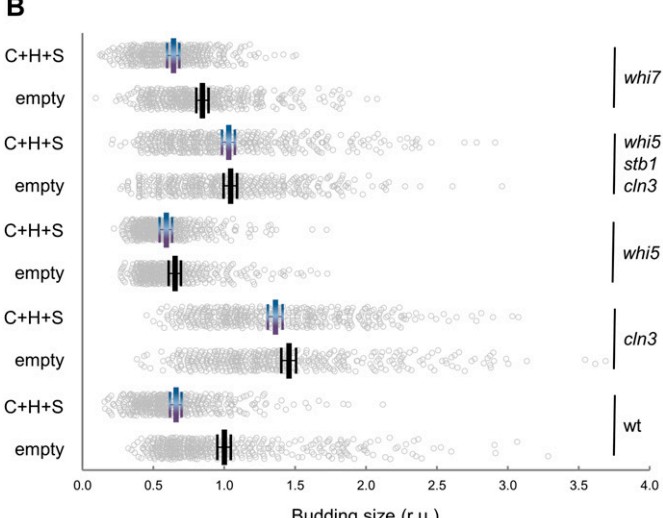

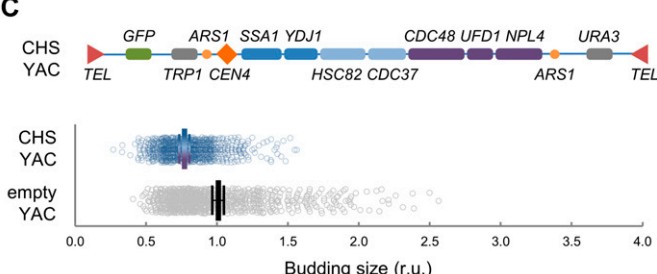

**Figure 2. Chaperones of the Hsp70, Hsp90, and Cdc48 systems are limiting for cell-cycle entry.**
**(A)** Budding volume of asynchronously growing cells transformed with the indicated combinations of compatible centromeric vectors encoding chaperones of the Hsp70 (S, *SSA1*, and *YDJ1*), Hsp90 (H, *HSC82*, and *CDC37*), and Cdc48 (C, *CDC48*, *UFD1*, and *NPL4*) systems. Individual budding volumes were determined and made relative to the mean value for wild-type cells transformed with the corresponding empty vectors. Mean values (N > 200) and confidence limits ($\alpha$ = 0.05, thin vertical lines) for the mean are plotted. **(B)** Cells with indicated genotypes were transformed with empty or compatible centromeric vectors encoding the three chaperone systems (C+H+S) and budding volumes of newly born cells were determined as in panel (A). Individual data (N > 400), mean values (thick vertical lines), and confidence limits ($\alpha$ = 0.05, thin vertical lines) for the mean are plotted. **(C)** Cells transformed with an artificial chromosome encoding the three chaperone systems (CHS YAC) or empty vector. Budding volumes of newly born cells were determined as in panel (A). Individual data (N < 400), mean values (thick vertical lines), and confidence limits ($\alpha$ = 0.05, thin vertical lines) for the mean are plotted.
Source data are available for this figure.

## Molecular competition for chaperones predicts cell size dependency on growth rate

As they are associated with client proteins mostly in a transient manner, the level of free chaperones might be inversely dependent on protein synthesis and trafficking rates, thus constituting a simple mechanism to report growth kinetics to the Start network and, hence, modulate cell size as a function of growth rate. To test this notion, we developed a mathematical model (Fig 3A) wherein protein synthesis and G1 Cdk–cyclin complex assembly compete for limiting amounts of Ydj1, the best characterized chaperone in terms of regulating Cln3. Ydj1 plays an activating role by releasing Cln3 from the ER during G1 (Vergés et al, 2007), but is also important for efficient degradation of Cln3 by Cdc28-dependent and autoactivated phosphorylation (Yaglom et al, 1996, 1995). Taken together, these data point to the notion that Ssa1/Ydj1 chaperones contribute to both proper Cln3 folding (i.e., binding Cdc28 in a productive conformation) and its release from the ER where the segregase Cdc48 also plays a key role (Parisi et al, 2018). Because these regulatory steps are likely related at the molecular level, we opted for treating them as a single event in the competition model. On another point, although the specific affinity of Ydj1 for the various Cln3 domains was similar to other proteins (Fig S4), the number of client proteins being synthesized at any given time is in overwhelming excess relative to those of Ydj1 (Gong et al, 2009) and especially of Cln3, which is present at very low levels throughout the G1 phase (Tyers et al, 1993; Cross et al, 2002).

In our model, the level of available, unbound Ydj1 (Ydj1$_A$) is a key variable and, because client-engaged chaperones offer a reduced mobility (Lajoie et al, 2012; Saarikangas et al, 2017), we decided to use FLIP and fluorescence correlation spectroscopy (FCS) to obtain a mobility index of fully functional Ydj1 and Ssa1 fusions to GFP as reporter of their availability in single cells. First, we tested the validity of this methodological approach by compromising chaperone function with AZC, which induces misfolded protein accumulation with chaperones into disperse cellular aggregates (Escusa-Toret et al, 2013), thus reducing levels of soluble available chaperones. Although GFP mobility remained unaffected, both Ssa1 and Ydj1 fusions to GFP decreased their mobility upon AZC treatment (Fig S5). These data support the use of mobility data as a proxy of chaperone availability in vivo.

Individual genetically identical cells display a large variability in multifactorial processes such as gene expression and growth (Blake et al, 2003; Ferrezuelo et al, 2012) and we reasoned that if the molecular competition model is correct, endogenous variability in growth rate should have an effect on chaperone availability at the single-cell level. We found that Ydj1-GFP diffusion was correlated negatively with growth rate in single G1 cells by both FLIP (Fig 3C and D) and FCS (Fig 3E and F) analysis and at a population level in media sustaining different growth rates (Fig S6A). The chaperone competition model perfectly fitted the dependence of Ydj1 availability on growth rate (Fig 3F) and, more important, it also simulated very closely the increase of the budding volume with growth rate (Ferrezuelo et al, 2012) (Fig 3G). As expected from a competition framework, the fitted parameter sets produced negative correlations between available Ydj1 and

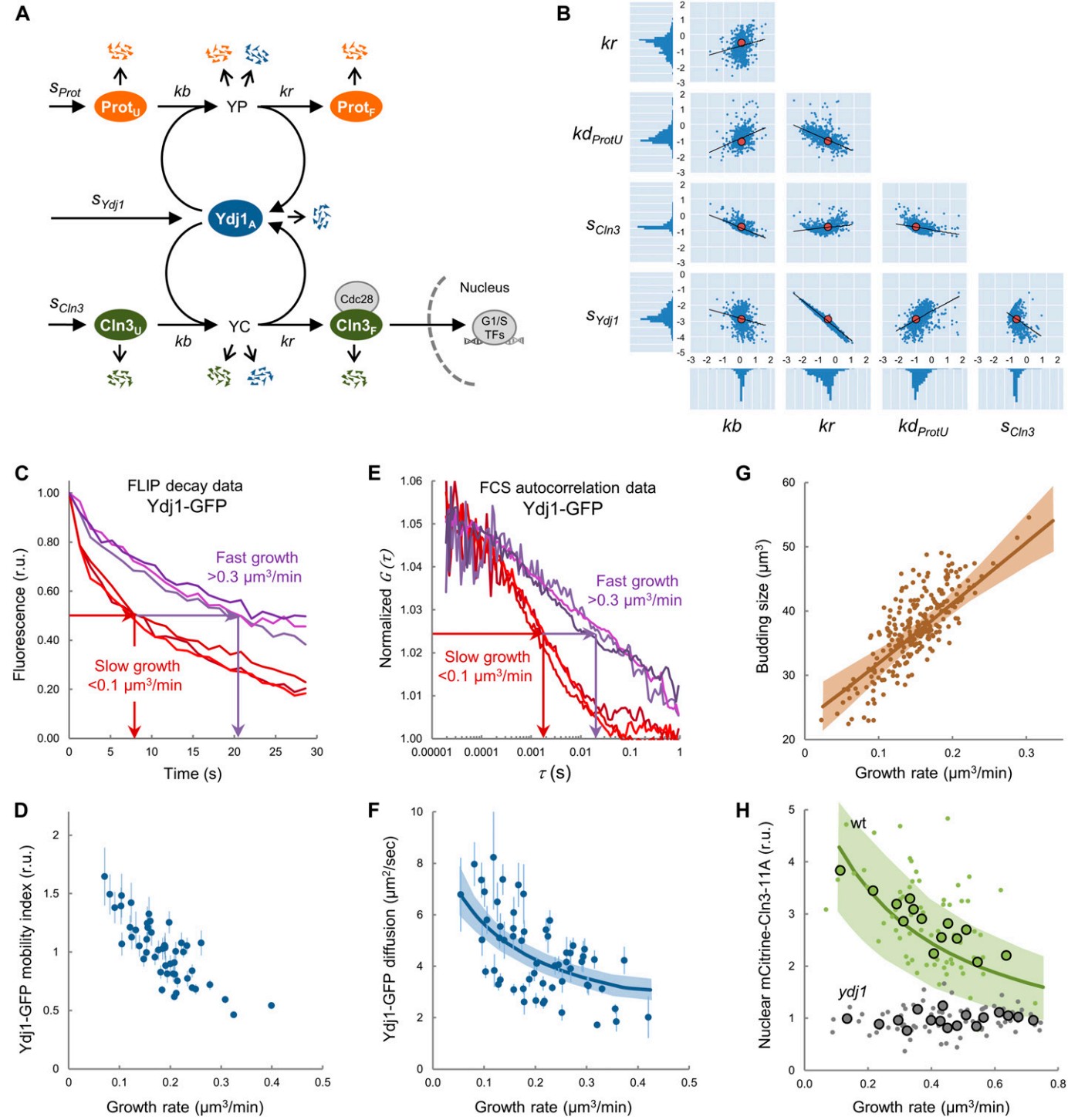

**Figure 3. The molecular competition model, chaperone availability, and nuclear accumulation of Cln3.**
**(A)** Scheme of the chaperone competition model connecting protein synthesis and Cln3 folding and complex formation with Cdc28. Chaperones associate to the unfolded protein ($Prot_U$) mostly in a transient manner until the properly folded protein ($Prot_F$) is released. Thus, protein synthesis rate would be a key determinant of the level of available Ydj1 ($Ydj1_A$) in the free pool and, as it is essential in proper folding and release of Cln3 ($Cln3_F$), it would in turn govern the rate of Whi5 phosphorylation in the nucleus for triggering Start. Variables and parameters used in the model are indicated. All key components of the competition model are subject to specific rates of degradation (open arrows) as described in the Materials and Methods section. **(B)** Distribution of parameters of the model fitted to experimental data of Ydj1 diffusion and critical volume as a function of growth rate. Parameter sets sampled using Markov Chain Monte Carlo (small blue circles) are plotted in $log_{10}$ space as well as the corresponding mode values (large red circles) and regression lines. One-dimensional histograms of parameter values are also plotted adjacent to the axes. **(C)** Cells expressing Ydj1-GFP were analyzed by FLIP in time-lapse experiments to determine also growth rate at a single-cell level. Individual FLIP decay data are plotted for three fast (>0.3 $\mu m^3$/min, purple lines) and three slow (<0.1 $\mu m^3$/min, red lines) growing cells. **(D)** Ydj1-GFP mobility indexes (circles) were obtained from FLIP decay curves as

unfolded target proteins (Fig S6B), and a positive correlation between the level of Ydj1 in complexes with either Cln3 or all other proteins (Fig S6C). Interestingly, the competition model produced acceptable fits with parameters spanning several orders of magnitude (Fig 3B), which underlines the robustness of the chaperone competition design in predicting growth rate–dependent chaperone availability and cell size, letting parameters to be adapted for subjugating Start to cellular processes other than growth. In summary, our experimental results and modeling simulations support the notion that growth rate modulates levels of available chaperones.

### Growth rate and protein synthesis modulate accumulation of Cln3 in the nucleus

The molecular competition model predicted that increasing growth rate would decrease available levels of free chaperones (Ydj1$_A$) and, hence, decrease the steady-state level of folded free Cln3 (Cln3$_F$) (Fig S6D and E). More precisely, the fraction of Ydj1 bound to client proteins (YP+YC) increases at higher growth rates. As a consequence, the fraction of available free Ydj1 (Ydj1$_A$) drops and the rate at which proteins can be folded is reduced, which in turn increases the fraction of unfolded proteins (Prot$_U$ and Cln3$_U$). Because of its intrinsic instability, folded Cln3 (Cln3$_F$) levels are extremely more sensitive to the effects of Ydj1 compared with all other proteins. Indeed, whereas the total concentration of mCitrine-Cln3-11A in G1 remained unaltered on average at different growth rates (Fig S6G), nuclear levels of mCitrine-Cln3-11A displayed a significant negative correlation ($P = 2 \times 10^{-4}$) with growth rate very similar to that predicted by the model (Fig 3H). Notably, this negative correlation was totally lost in Ydj1-deficient cells (Fig 3H). In summary, higher growth rates reduce the nuclear levels of Cln3 at a given volume in a chaperone-dependent manner.

Next, with the purpose of simulating lower growth rates, we used cycloheximide (CHX) to decrease protein synthesis and relieve chaperones temporarily from the load of nascent polypeptides (Fig 4A and B). As expected, CHX inhibited incorporation of puromycin in cell-free extracts (Fig 4C), but it increased the rate at which heat-treated luciferase was renatured and became active, indicating that both protein refolding and synthesis compete for chaperones in cell extracts. Next, we tested the effects of CHX at 0.2 µg/ml, a sublethal concentration that does not activate the environmental stress response (Jacquet, 2003; Trotter et al, 2002), and found that the protein synthesis rate displayed a 5.9-fold decrease in live cells (Fig S7). In agreement with our model prediction (Fig 4B), this sublethal dose of CHX increased the average diffusion coefficient of Ydj1-GFP as

measured by FCS (Fig 4D) and FLIP (Fig 4E). Temporary perturbations of chaperone availability should in turn have an effect in the nuclear accumulation of Cln3 (Fig 4B). To quantify endogenously expressed Cln3-3HA in immunofluorescence images, we used a semi-automated method that analyzes both cytoplasmic and nuclear compartments in fixed yeast cells (Yahya et al, 2014). Notably, we found that relative nuclear levels of Cln3-3HA rapidly increased after addition of CHX (Fig 4F and G), decreasing at later times as predicted by the model (Fig 4B), and this transient increase fully depended on Ydj1. However, CHX has been shown to increase cell size at Start (Jorgensen et al, 2004). This apparent discrepancy could originate from the different short- and long-term effects of CHX, that is, increasing Ydj1 mobility and nuclear localization of Cln3 at very short times (less than 1 min), but eventually decreasing G1 cyclin levels, which is what would finally result in a larger cell size. Overall, these data support the notion that chaperone availability transmits growth and protein synthesis rate information to modulate the rate at which the G1 Cdk–cyclin complex is properly formed and accumulates in the nucleus.

### Stress conditions that decrease chaperone availability prevent accumulation of Cln3 in the nucleus

Yeast cells respond to stress conditions as diverse as high temperature, high osmolarity, or abnormal levels of unfolded proteins in the ER, by highly conserved transcriptional programs that increase chaperone expression to protect damaged proteins from aggregation, unfold aggregated proteins, and refold damaged proteins or target them for efficient degradation (De Nadal et al, 2011). Thus, stresses are assumed to cause a temporary deficit in chaperone availability. Hsf1, the key transcriptional activator of the heat shock response, is inhibited by chaperones of the Hsp70 and Hsp90 systems, and it has been proposed that the accumulation of unfolded or damaged proteins would readily titrate the chaperone machinery from Hsf1, allowing derepression of the transcription factor (Verghese et al, 2012). Supporting this notion, Ydj1-assisted Ssa1 chaperone is targeted to and accumulates in protein aggregates (Mogk et al, 2018) after heat stress. A similar titration mechanism has been proposed for Kar2, the Hsp70 chaperone acting at the ER lumen (Gardner et al, 2013). On the other hand, both heat and salt stress have been shown to inhibit the G1/S regulon (Rowley et al, 1993; Bellí et al, 2001; Trotter et al, 2001). Thus, we reasoned that chaperone titration by stress could effectively reduce chaperone availability and restrain nuclear accumulation of Cln3 in a temporary manner (Fig 5A). We first interrogated our model and simulated a stress event by transferring different fractions of

shown in panel (C) and plotted as a function of growth rate with the corresponding confidence limits ($\alpha = 0.05$). **(E)** Cells expressing Ydj1-GFP were analyzed by FCS in time-lapse experiments to determine also growth rate at a single-cell level. Individual FCS autocorrelation data are plotted for three fast (>0.3 µm$^3$/min, purple lines) and three slow (<0.1 µm$^3$/min, red lines) growing cells. **(F)** Ydj1-GFP diffusion coefficients (circles) were obtained from FCS autocorrelation functions as shown in panel (E) and plotted as a function of growth rate with the corresponding standard error. The mean fit produced by the full ensemble of parameter sets shown in panel (B) is plotted as a line with one SD intervals. **(G)** The chaperone competition model predicts the critical size being a function of growth rate at the single-cell level. Experimental budding volumes as a function of growth rate (closed circles) and the mean fit as in panel (F) are shown. **(H)** Nuclear to cytoplasmic ratios for mCitrine-Cln3-11A from G1 wt (green, N = 68) and Ydj1-deficient (*ydj1*, gray, N = 85) cells as a function of growth rate. Mean values (large circles) and confidence limits ($\alpha = 0.05$) for binned (5 cells/bin) data are also shown. A simulation of Cln3F was obtained with the parameter set 3114 within a fourfold range of *kd* and *kr*, and the resulting mean (green line) is plotted with one SD intervals.
Source data are available for this figure.

**Figure 4. Protein synthesis is a key determinant of chaperone availability and nuclear accumulation of Cln3.**
**(A)** The competition model predicts that reducing the protein synthesis rate would decrease the requirements of Ydj1 chaperone in folding of new proteins (Prot$_F$) and increase the level of available chaperone (Ydj1$_A$) and free Cln3 (Cln3$_F$) levels as a function of time after CHX addition. **(B)** Prediction of available chaperone (Ydj1$_A$) and free Cln3 (Cln3$_F$) levels as a function of time after CHX addition. Simulations were produced by using the parameter set 3114 and varying (3, 4.5, 6, 9, or 12-fold) reductions in protein synthesis rates around the experimental value (Fig S7). **(C)** Refolded luciferase activity as a function of time in yeast cell extracts treated or not with 20 μg/ml CHX in the presence of an ATP-regenerating system. Inset: immunoblot analysis of puromycin incorporation in cell-free extracts used for refolding analysis in the absence or presence of 20 μg/ml CHX. Numbers refer to relative incorporation levels as measured by densitometric analysis. **(D)** Cells expressing Ydj1-GFP were analyzed by FCS before (−) or 5 to 10 min after adding a sublethal dose of CHX at 0.2 μg/ml (+). Individual protein diffusion coefficients are plotted (N > 50). Mean values (thick lines) and confidence limits for the mean (α = 0.05, thin lines) are also shown. **(E)** Ydj1-GFP and GFP mobility assayed by FLIP at the indicated time points after CHX addition at 0.2 μg/ml. Relative mean values and confidence limits (α = 0.05) for the mean are shown. **(F)** Nuclear levels of Cln3-3HA by immunofluorescence before or 60 s after addition of CHX at 0.2 μg/ml. **(G)** Nuclear accumulation of Cln3-3HA in asynchronous individual wt and Ydj1-deficient (*ydj1*) cells before or at the indicated times after CHX addition as in panel (F). Relative mean values (N > 200) and confidence limits (α = 0.05, thin horizontal lines) for the mean are shown.
Source data are available for this figure.

the folded protein (Prot$_F$) to the unfolded population (Prot$_U$). As shown in Fig 5B, the model predicted a sharp and transient reduction in available Ydj1 (Ydj1$_A$) and free Cln3 (Cln3$_F$). Notably, we found that Ydj1-GFP diffusion decreased very rapidly under both heat and salt stress (Fig 5C and D) and recovered at later times to attain similar steady states to the prestress situation. Moreover, nuclear levels of Cln3-3HA displayed a similar and transient decrease after heat and salt stress (Fig 5E and F), which did not affect overall Cln3-3HA levels as measured by immunoblotting. Because of its extremely short half-life, Cln3 is thought to respond very rapidly to new conditions. However, nuclear levels of Cln3 only recovered prestress values after 20–30 min (Fig 5G and H), within the same time range needed by the Ydj1 chaperone to recover prestress mobility (Fig 5C and D), thus reinforcing a functional link between chaperone availability and nuclear accumulation of Cln3 under stress conditions. To test this further, we used our model to predict the behavior of Cln3 during stress adaptation, assuming

that overall protein folding and nuclear accumulation of Cln3 use chaperones in independent or competing manners. As seen in Fig 5G and H, only the competition scenario was able to recapitulate the Cln3 immunofluorescence data from both heat and salt stresses.

ER stress causes protein aggregation in the cytoplasm (Hamdan et al, 2017), and increases Ssa4 expression to levels very similar to Kar2 (Travers et al, 2000), suggesting that ER stress also affects Ydj1 availability. Moreover, ER stress also causes a G1 delay (Vai et al, 1987). We found that similar to heat and salt stress, addition of tunicamycin to induce irreversible ER stress in yeast cells decreased Ydj1 mobility and nuclear levels of Cln3-3HA (Fig S8). Overall, these data indicate that stress conditions due to very different environmental cues cause coincident decreases in Ydj1 mobility and Cln3 nuclear accumulation, reinforcing the notion that chaperone availability is a key parameter that controls G1 cyclin fate and adapts cell-cycle entry to growth and stress.

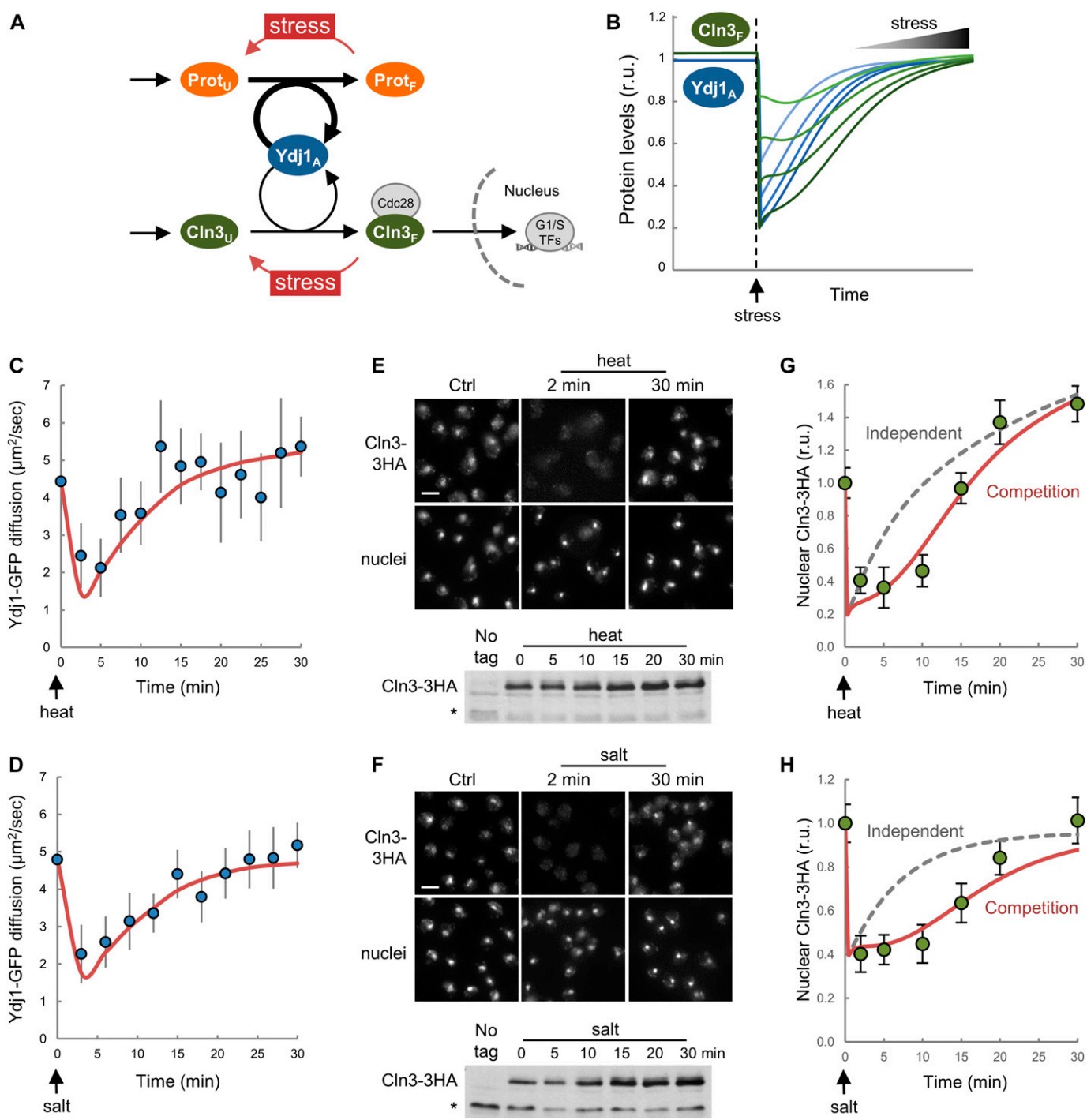

**Figure 5. Stress reduces chaperone mobility and hinders nuclear accumulation of Cln3.**
**(A)** The competition model predicts that a sudden increase in the level of unfolded proteins (Prot$_U$) would decrease the level of available chaperone (Ydj1$_A$) and, hence, reduce the level of free Cln3 (Cln3$_F$). **(B)** Prediction of available chaperone (Ydj1$_A$) and free Cln3 (Cln3$_F$) levels as a function of time after stress. Simulations were produced by using the parameter set 3114 and transferring different fractions (20–80%) of folded protein (Prot$_F$) to the unfolded population (Prot$_U$). **(C, D)** Mean Ydj1-GFP diffusion coefficients (N > 20, filled circles) and confidence limits ($\alpha$ = 0.05, thin lines) are plotted at different times during heat (C) or salt (D) stress. Model fits (see text for details) are also shown (orange lines). **(E, F)** Immunofluorescence of Cln3-3HA in late-G1 cells arrested with $\alpha$ factor during heat (E) and salt (F) stress. Bottom: Cln3-3HA levels by immunoblotting. A cross-reacting band is shown as loading control. **(G, H)** Nuclear levels of Cln3-3HA quantified from cells during heat (G) and salt (H) stress as in panels (E) and (F). Relative mean values (N > 200) and confidence limits ($\alpha$ = 0.05, thin horizontal lines) for the mean are shown. Model simulations (see text for details) assuming that ProtU and Cln3U require Ydj1A in independent (gray dashed lines) or competing (solid orange lines) scenarios are also shown.
Source data are available for this figure.

# Discussion

Our data show that multifunctional chaperone systems play a limiting role in driving nuclear accumulation of the Cdc28–Cln3 complex during G1 progression. We also show that increased growth rates, by allocating higher levels of chaperones to protein synthesis and trafficking, reduce free chaperone pools and restrain nuclear accumulation of Cln3 at a given volume in G1. If Cln3 has to reach a threshold concentration (Wang et al, 2009; Liu et al, 2015) set by Whi5 dilution during G1 to execute Start (Schmoller et al, 2015), fast-growing cells would delay G1 progression to grow larger and attain the same pool of free chaperones and nuclear Cln3 (Fig 6A). As a consequence, the critical size would be set as a function of growth rate. Because of the close molecular connection proposed here between protein synthesis and folding/release of the G1 cyclin, our model would allow cells to quickly adjust their size to many

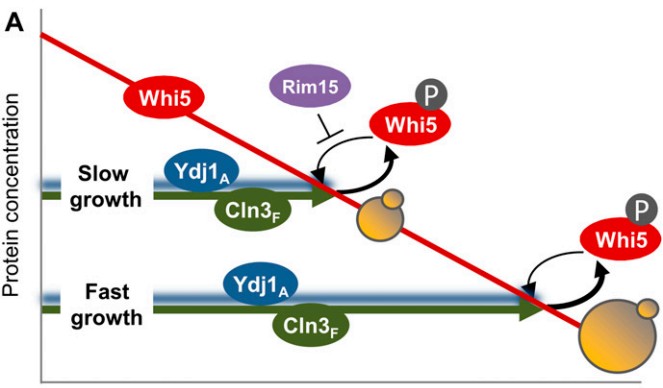

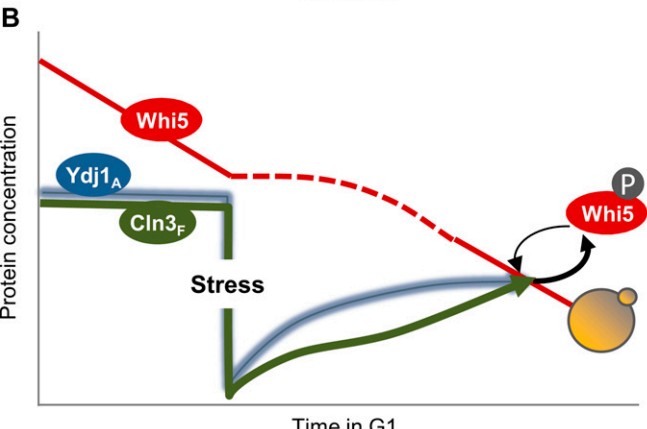

**Figure 6.  Competition for chaperones and the regulation of cell-cycle entry by growth and stress.**
Chaperone-dependent Cln3 folding and release would act as a key modulator of Start to subordinate cell-cycle entry to growth and stress. **(A)** By compromising higher levels of chaperones in growth processes, fast-growing cells exhibit lower levels of available chaperones (Ydj1$_A$) and restrain accumulation of Cln3 in the nucleus (Cln3$_F$), thus delaying Start until a larger cell size is attained and Whi5 is diluted to lower levels by growth (Schmoller et al, 2015). In addition, when cells grow slower because of poor carbon sources, Rim15 counteracts Whi5 dephosphorylation in G1 to facilitate cell-cycle entry (Talarek et al, 2017). **(B)** On the other hand, stressful conditions compromise chaperone availability and delay Start until normal proteostasis conditions are restored and Cln3 recovers proper levels in the nucleus to counteract Whi5.

different intrinsic and extrinsic signals as long as they modulate protein synthesis, thus acting as a common mediator of specific growth signaling pathways and the Start network. This view would also explain, at least in part, why deleterious mutations in nutrient sensing pathways that control ribosome biogenesis and cell growth cause a small cell size phenotype (Baroni et al, 1994; Tokiwa et al, 1994; Jorgensen et al, 2004). A recent comprehensive analysis of cell size mutants concluded that in most *whi* mutants, the small cell size was due to indirect effects mostly caused by a decrease in growth rate (Soifer & Barkai, 2014).

The chaperone competition mechanism would only operate above a minimal translation rate to sustain G1–cyclin synthesis (Schneider et al, 2004) and would collaborate with pathways regulating G1–cyclin expression by specific nutrients (Baroni et al, 1994; Tokiwa et al, 1994; Gallego et al, 1997; Newcomb et al, 2002). In this respect, Cln3 synthesis is strongly down-regulated at low growth rates by a uORF-dependent mechanism (Polymenis & Schmidt, 1997), shifting cells from poor to rich carbon sources increases Cln3 abundance but yet delays Start in terms of cell size (Johnston et al, 1979; Tokiwa et al, 1994; Hall et al, 1998), phosphate deprivation increases degradation of Cln3 by Pho85-dependent phosphorylation (Menoyo et al, 2013), and Rim15 counteracts Whi5 dephosphorylation in G1 to facilitate cell-cycle entry in poor carbon sources (Talarek et al, 2017). Moreover, nutrient modulation of cell size is largely independent of the core components of the Start network (Jorgensen et al, 2004). From this point of view, modulation of the critical cell size threshold by nutrients and growth rate has remained a mysterious issue during decades (Fantes & Nurse, 1977; Johnston et al, 1979; Tyson et al, 1979). Our findings provide a general mechanism to modulate nuclear accumulation of Cln3 and cell-cycle entry as a function of growth rate (Fig 6A).

Cells deficient for Cln3 still maintain a growth rate–dependent size at Start, but with a much larger variability compared with wild-type cells (Yahya et al, 2014). Cln1 and Cln2 become essential in the absence of Cln3, and cells lacking these two other G1 cyclins also display a wider range of sizes at Start as a function of growth rate. Moreover, the *CLN2* mRNA has also been found enriched in Whi3 pulldowns (Colomina et al, 2008; Holmes et al, 2013), suggesting that the chaperone competition mechanism would also apply to Cln2, particularly in the absence of Cln3.

Whi5 levels have been found to decrease in rich carbon sources (Liu et al, 2015). If this modulation was responsible for cell size adaptation, we should expect cells displaying the opposite behavior, that is, cells should be smaller in rich carbon sources. However, this observation could be interpreted as an effect, rather than a causative determinant of cell size adaptation. Whi5 is synthesized in a size-independent manner (Schmoller et al, 2015) and, also likely, in a growth rate–independent manner. As cells growing in rich carbon sources are larger, this effect would produce a decrease in Whi5 concentration. Thus, although Whi5 would act as a growth rate–independent sizer, chaperone availability would dynamically transmit growth rate information to modulate G1 Cdk activation and adjust cell size as a function of the individual growth potential.

Available models of the cell cycle of budding yeast have centered their attention to different aspects of the molecular machineries that execute and regulate key transitions, but the implications of

growth rate on cell size have been addressed only in a few occasions. Thus, a model based on intrinsic size homeostasis predicts growth rate dependence if nutrient uptake is subject to geometric constraints (Spiesser et al, 2012, 2015). In this regard, surface-to-volume ratios have been recently highlighted in bacteria as key parameters for setting cell size as a function of growth rate (Harris & Theriot, 2016). Interestingly, an important fraction of Ydj1 is involved in post-translational protein translocation at the ER (Caplan et al, 1992; McClellan et al, 1998) to fuel cell surface growth which, as a consequence, would set Ydj1 demands as a function of surface-to-volume ratios. On the other hand, in a recent model of the G1/S transition, it has been proposed that growth rate dependence would be exerted, directly or indirectly, by ribosome biogenesis effects on the Cln3/Whi5 interplay (Palumbo et al, 2016). In agreement with this idea, here, we propose that chaperones act as key mediators in this pathway by modulating the ability of Cln3 to accumulate in the nucleus and, hence, attain a critical Cln3/Whi5 ratio.

Nutrient modulation of cell size also takes place during bud growth and, as we had observed in G1, cell size at birth displays a strong correlation with growth rate at a single-cell level during the preceding S–G2–M phases (Leitao & Kellogg, 2017). Thus, mechanisms sensing growth rate independently of the specific nutritional conditions also operate during bud volume growth.

Finally, we show that different stress agents cause concurrent decreases in Ydj1 mobility and nuclear Cln3 levels, supporting the idea that chaperone availability is a key factor controlling localization of the most upstream G1 cyclin and, hence, modulating G1 length under stress conditions. Heat shock transiently inhibits G1/S gene expression (Rowley et al, 1993), but the molecular mechanism is still unknown. On the other hand, osmotic shock causes a similar temporary repression of the G1/S regulon (Bellí et al, 2001), where Hog1-mediated phosphorylation of Whi5 and Msa1 plays an important role in transcription inhibition (González-Novo et al, 2015). However, G1/S gene expression was still repressed by salt in the absence of Whi5 and Msa1, indicating the existence of additional mechanisms sufficient to inhibit the G1/S regulon under osmotic shock. It has long been known that ER stress causes a G1 delay (Vai et al, 1987), but the molecular mechanisms have not been described. Our results uncover a new mechanism that would explain the observed rapid down-regulation of G1/S genes by heat, salt, and ER stress, whereby immediate titration of chaperones would decrease available pools of Ssa1 and Ydj1 chaperones required to accumulate the G1 Cdk in the nucleus for triggering Start (Fig 6B). At later times after stress, Msn2,4-dependent up-regulation of Cip1 would contribute to inhibit G1 Cdk activity (Chang et al, 2017). In summary, although stress conditions would activate different mechanisms contributing to delay G1 progression, we propose that alterations in proteostasis and chaperone availability would be the earliest and most direct effectors leading to an abrupt G1 arrest by stress.

Free chaperone levels could also report growth capability to other processes influenced by growth rate (Brauer et al, 2008) or stressful conditions (De Nadal et al, 2011) through similar competition settings. Furthermore, imposing a high level of free chaperones as a requirement for Start would ensure their availability in highly demanding downstream processes such as polarized growth for bud emergence or nucleosome remodeling during replication.

# Materials and Methods

### Strains and plasmids

Yeast strains and plasmids used are listed in Table S1. Parental strains and methods used for chromosomal gene transplacement and PCR-based directed mutagenesis have been described (Gallego et al, 1997; Ferrezuelo et al, 2012). Unless stated otherwise, all gene fusions in this study were expressed at endogenous levels at their respective loci. As C-terminal fusions of GFP or other tags has strong deleterious effects on Ydj1 function (Saarikangas et al, 2017), we inserted GFP at amino acid 387, between the dimerization domain and the C-terminal farnesylation sequence of Ydj1. This construct had no detectable effects on growth rate or cell volume when expressed at endogenous levels. The mCitrine-Cln3-11A fusion protein contains a hypoactive and hyperstable cyclin with 11 amino acid substitutions (R108A, T420A, S449A, T455A, S462A, S464A, S468A, T478A, S514A, T517A, and T520A) that allows its detection by fluorescence microscopy with no gross effects on cell-cycle progression (Schmoller et al, 2015).

### Growth conditions

Cells were grown for 7–8 generations in SC medium with 2% glucose at 30°C before microscopy unless stated otherwise. Other carbon sources used were 2% galactose, 2% raffinose, and 3% ethanol. GAL1p-driven gene expression was induced by addition of 2% galactose to cultures grown in 2% raffinose at OD600 = 0.5. When stated, 1 µM β-estradiol was used to induce the GAL1 promoter in strains expressing the Gal4-hER-VP16 (GEV) transactivator (Louvion et al, 1993). AZC was used at 10 mM, and CHX was added at a sublethal dose of 0.2 µg/ml that does not trigger stress gene activation (Trotter et al, 2002; Jacquet, 2003). Heat and osmotic stresses were imposed by transferring cells from 25°C to 37°C or adding 0.4 M NaCl at 30°C, respectively. Tunicamycin was added used at 1 µg/ml. Small newly born cells were isolated from Ficoll gradients (Mitchison, 1988).

### Time-lapse microscopy

Cells were analyzed by time-lapse microscopy in 35-mm glass-bottom culture dishes (GWST-3522; WillCo) in SC-based media at 30°C essentially as described (Ferrezuelo et al, 2012) using a fully motorized Leica AF7000 microscope. Time-lapse images were analyzed with the aid of BudJ (Ferrezuelo et al, 2012), an ImageJ (Wayne Rasband, NIH) plugin that can be obtained from http://www.ibmb.csic.es/groups/spatial-control-of-cell-cycle-entry to obtain cell dimensions and fluorescence data. Volume growth rate in G1 were obtained as described (Ferrezuelo et al, 2012). Start was estimated at a single-cell level as the time where the nuclear to cytoplasmic ratio of Whi5 had decreased below 1.5. Photobleaching during

acquisition was negligible (less than 0.1% per time point), and autofluorescence from naive cells was always subtracted.

## Determination of nuclear and cellular concentrations of fluorescent fusion proteins

Wide-field microscopy is able to collect the total fluorescence emitted by yeast cells and, consequently, cellular concentration of fluorescent fusion proteins was obtained by dividing the integrated fluorescence signal within the projected area of the cell by its volume. Regarding the quantification of nuclear levels, because the signal in the nuclear projected area is influenced by both nuclear and cytoplasmic fluorescence, determination of the nuclear concentration required specific calculations as described in Fig S1A. In confocal microscopy, the fluorescence signal is directly proportional to the concentration of the fluorescent fusion protein, and required no further calculations. The nuclear compartment was delimited as described (Ferrezuelo et al, 2012).

## Chaperone mobility analysis by FLIP and FCS

We used FLIP and FCS to analyze chaperone mobility in a Zeiss LSM780 confocal microscope. FLIP was used as a qualitative assay to determine Ssa1-GFP and Ydj1-GFP mobility in the whole cell. A small circular region of the cytoplasm (3.6 $\mu m^2$) was repetitively photobleached at full laser power, whereas the cell was imaged at low intensity every 0.5 s to record fluorescence loss. After background subtraction, fluorescence data were corrected to discard the small loss of fluorescence caused by image acquisition as measured in control cells in the same field that were not purposely bleached. Then, fluorescence data from an unbleached cell region were made relative to the initial time point, and a mobility index was calculated as the inverse of the fluorescence half-life obtained by fitting an exponential function to fluorescence emitted as a function of time. Quantitative analysis of Ydj1-GFP diffusion by FCS was performed essentially as described (Saarikangas et al, 2017). Specifically, FCS analysis was performed at 25°C to minimize signal variability in the 0.1–1-s range, and cells were prebleached to attain count rates within the 50–100 kHz range during acquisition for periods of 5 s. FCS correlation data were fitted in the 10 μsec to 100 msec range of time intervals with the aid of QuickFit 3 (http://www.dkfz.de/Macromol/quickfit/), assuming a 1-component anomalous mode of diffusion ($\alpha$ = 0.5) in the Levenberg–Marquardt algorithm to obtain diffusion coefficients. Duplicate measurements were always taken, and outliers were removed from analysis if the relative standard error of the fitted coefficient of diffusion was higher than 50% or the fitted autocorrelation intersect was higher than 1.01 as a result of strong perturbations in the average count rate during acquisition. In time-lapse experiments, outliers were removed if the

relative difference to neighbor values was higher than 50%. Removed outliers were always less than 5% of measurements.

## Nuclear import rate determinations by FLIP

To analyze nuclear import kinetics of Cdc28-GFP, a circle inscribed within the Htb2-mCherry nuclear region was repetitively photobleached, whereas the cell was imaged every 0.5 s to record fluorescence loss. After background subtraction, fluorescence data were corrected as above. Finally, fluorescence signals within nuclear and cytoplasmic areas were used to analyze import kinetics as described in Fig S2A. The export rate was assumed constant among G1 cells and obtained as described in Fig S2B.

## Immunofluorescence

Immunofluorescence of endogenous levels of Cln3-3HA was carried out by a signal amplification method (Vergés et al, 2007) with αHA (clone 3F10; Roche) and goat-αrat and rabbit-αgoat Alexa555-labeled antibodies (Molecular Probes) on methanol-pemeabilized cells. To analyze localization of Cln3-3HA, we used N2CJ, a plugin for ImageJ (Wayne Rasband, NIH), to perform accurate quantification in both cytoplasmic and nuclear compartments of cells (Yahya et al, 2014).

## Model equations

The model in Fig 3A was simulated with a set of nonlinear differential equations.

$$\frac{d\textbf{\textit{Prot}}_\textbf{U}}{dt} = s_{Prot}*Vol - kb*\textbf{\textit{Prot}}_\textbf{U}*\frac{\textbf{\textit{Ydj}}1_\textbf{A}}{Vol} - kd_{ProtU}*\textbf{\textit{Prot}}_\textbf{U} + kd_{Ydj1}*\textbf{\textit{YP}} \quad (1)$$

$$\frac{d\textbf{\textit{YP}}}{dt} = kb*\textbf{\textit{Prot}}_\textbf{U}*\frac{\textbf{\textit{Ydj}}1_\textbf{A}}{Vol} - k_r*YP - kd_{Ydj1}*\textbf{\textit{YP}} - kd_{ProtU}*\textbf{\textit{YP}} \quad (2)$$

$$\frac{d\textbf{\textit{Prot}}_\textbf{F}}{dt} = kr*\textbf{\textit{YP}} - \beta*kd_{ProtU}*\textbf{\textit{Prot}}_\textbf{F} \quad (3)$$

$$\frac{d\textbf{\textit{Cln3}}_\textbf{U}}{dt} = s_{Cln3}*Vol - kb*\textbf{\textit{Cln3}}_\textbf{U}*\frac{\textbf{\textit{Ydj}}1_\textbf{A}}{Vol} - kd_{Cln3U}*\textbf{\textit{Cln3}}_\textbf{U} + kd_{Ydj1}*\textbf{\textit{YC}}$$
$$(4)$$

$$\frac{d\textbf{\textit{YC}}}{dt} = kb*\textbf{\textit{Cln3}}_\textbf{U}*\frac{\textbf{\textit{Ydj}}1_\textbf{A}}{Vol} - kr*\textbf{\textit{YC}} - kd_{Ydj1}*\textbf{\textit{YC}} - kd_{Cln3U}*\textbf{\textit{YC}} \quad (5)$$

$$\frac{d\textbf{\textit{Cln3}}_\textbf{F}}{dt} = kr*\textbf{\textit{YC}} - kd_{Cln3F}*\textbf{\textit{Cln3}}_\textbf{F} \quad (6)$$

$$\frac{d\textbf{\textit{Ydj}}1_\textbf{A}}{dt} = s_{Ydj1}*Vol - kb*\textbf{\textit{Prot}}_\textbf{U}*\frac{\textbf{\textit{Ydj}}1_\textbf{A}}{Vol} - kb*\textbf{\textit{Cln3}}_\textbf{U}*\frac{\textbf{\textit{Ydj}}1_\textbf{A}}{Vol} + kr*\textbf{\textit{YP}} + kr*\textbf{\textit{YC}} - kd_{Ydj1}*\textbf{\textit{Ydj}}1_\textbf{A} + kd_{Cln3U}*\textbf{\textit{YC}} + kd_{ProtU}*\textbf{\textit{YP}} \quad (7)$$

**Table 1.** Model parameters.

| Parameter | Definition | Type | Value | Parameter set 3114 |
|---|---|---|---|---|
| $Vol$ | Cell volume | Controlled | 10–100 fl | |
| $S_{ProtU}$ | Protein synthesis | Controlled | 0.01–1 molec·s$^{-1}$·fl$^{-1}$ | |
| $S_{Cln3}$ | Cln3 synthesis | Fitted | $4.9 \times 10^{-2}$–4.8 molec·s$^{-1}$·fl$^{-1}$ | 0.126 molec·s$^{-1}$·fl$^{-1}$ |
| $S_{Ydj1}$ | Ydj1 synthesis | Fitted | $4.8 \times 10^{-5}$–$6.0 \times 10^{-2}$ molec·s$^{-1}$·fl$^{-1}$ | $9.19 \times 10^{-4}$ molec·s$^{-1}$·fl$^{-1}$ |
| $kd_{ProtU}$ | ProtU degradation | Fitted | $6.4 \times 10^{-3}$–31.5 s$^{-1}$ | $7.26 \times 10^{-2}$ s$^{-1}$ |
| $kd_{ProtF}$ | ProtF degradation | Fitted | 0.01* $kd_{ProtU}$·s$^{-1}$ | $7.26 \times 10^{-4}$ s$^{-1}$ |
| $kb$ | Binding to Ydj1 | Fitted | $1.4 \times 10^{-2}$–26.7 fl·molec$^{-1}$·s$^{-1}$ | 0.891 fl·molec$^{-1}$·s$^{-1}$ |
| $kr$ | Release from Ydj1 | Fitted | $2.8 \times 10^{-3}$–7.7 s$^{-1}$ | 0.388 s$^{-1}$ |
| $kd_{Cln3U}$ | Cln3U degradation | Fixed | $6.93 \times 10^{-2}$ s$^{-1}$ | |
| $kd_{Cln3F}$ | Cln3F degradation | Fixed | $3.85 \times 10^{-2}$ s$^{-1}$ | |
| $kd_{Ydj1}$ | Ydj1 degradation | Fixed | $5.77 \times 10^{-3}$ s$^{-1}$ | |

This model has seven state variables: **Prot$_U$**, unfolded proteins; **YP**, Ydj1-bound proteins; **Prot$_F$**, folded proteins; **Cln3$_U$**, unfolded Cln3; **YC**, Ydj1-bound Cln3; **Cln3$_F$**, folded Cln3; and **Ydj1$_A$**, free available Ydj1.

## Model parameters and simulations

In the model, we define 11 parameters (Table 1). Ydj1 binding and release ($kb$ and $kr$), which are assumed to be the same for $Prot_U$ and $Cln3_U$ (see Fig S4), and synthesis ($s_{Prot}$, $s_{Cln3}$, and $s_{Ydj1}$) and degradation ($kd_{ProtU}$, $kd_{Cln3U}$, $kd_{Cln3F}$, and $kd_{Ydj1}$) of the different components. A scaling factor ($\beta = 0.01$) is used to ensure that folded protein ($Prot_F$) has a half-life 100 times that of unfolded protein ($Prot_U$). We assume that $Prot_U$ synthesis rate is proportional to the cell volume ($Vol$), as well as to the cell-specific volume growth rate ($S_{Vol}$), which can be expressed as $S_{Vol} = S_{Prot} * \gamma$, where $S_{Prot}$ is the cell-specific protein synthesis rate and $\gamma$ is a scale conversion factor. Because of lack of data and to avoid over-fitting, $\gamma$ was fixed at 1. Because $CLN3$ and $YDJ1$ mRNA levels relative to total mRNA are quite constant at different growth rates (Slavov & Botstein, 2011), we have kept their synthesis rate constant and independent of the individual specific growth rate. The state variables in the model are in units of molecular number, not concentration, and therefore all zero- and second-order reactions are explicitly scaled by cell volume ($Vol$). In the model, we assume that the rate of change of cell volume with time is much lower than the rates of the biochemical reactions studied. This allowed us to treat the cell volume as a pseudo-parameter, so the steady state of other variables with respect to cell volume and growth rate ($S_{ProtU}$) could be analyzed more straightforwardly. The half-life of unfolded ($kd_{Cln3U}$) Cln3 was set to be 1.8 times longer than folded ($kd_{Cln3F}$) Cln3 as deduced from steady-state levels of Cln3-3HA in wild-type and Ydj1-deficient cells (Yaglom et al, 1996), whereas Ydj1 is a stable protein, with a half-life 20 times longer than folded Cln3.

To reduce the degrees of freedom available for modeling, degradation rates of all molecules were kept constant and independent if they are in a complex or in free form. The underlying reason is that molecules should be still recognized by their respective degradation machinery independent of their binding partners. We converted half-lives to degradation rates using the formula $\lambda = \log(2)/t_{1/2}$. The remaining five parameters were used to quantitatively fit two sets of measurements: the relationship between the diffusion rate of Ydj1 and growth rate in G1 cells (Fig 3F) and the relationship between budding volume and growth rate (Fig 3G).

## Budding volume versus growth rate

In the model, we assume that Start is triggered by Cln3 when it reaches a given critical threshold ($Cln3_F^{crit} = 25$) that together with Cdc28, it is needed to phosphorylate and inactivate a fixed given amount of DNA-bound Whi5/SBF complexes (Wang et al, 2009; Schmoller et al, 2015). Our objective was then to minimize the difference between experimental measurements and simulation results by changing the five free parameters in $L1(\theta) = (Vol(s_{Vol})_{exp} - Vol(\theta \mid s_{Vol})_{sim})^2$, where $Vol (S_{Vol})_{exp}$ is the experimentally measured budding volume (as a proxy for the critical volume at Start) of a cell growing at rate $S_{Vol}$ and $Vol(\theta \mid s_{Vol})_{sim}$ is the volume of cells growing at rate $S_{Vol}$ and reach a steady-state value where $Cln3_F$ equals the critical threshold $Cln3_F^{crit}$.

## Ydj1 diffusion versus growth rate

We assume that Ydj1 is present in two distinct pools: a fast diffusing free fraction ($Ydj1_A$) and a slowly diffusing fraction bound to either $Prot$ or $Cln3$ ($YP+YC$). To compare the experimentally measured diffusion rate of Ydj1 to our simulations, we define the simulated diffusion rate of Ydj1 as the weighted average diffusion coefficient for all species of Ydj1 in our model as follows:

$$D_{sim} = \frac{Ydj1_A}{Ydj1} * D_{free} + \frac{YP + YC}{Ydj1} * D_{bound}. \tag{8}$$

For Ydj1-GFP, we estimated that $D_{bound} = 1 \; \mu m^2/min$ from minimal values observed in AZC-treated cells and $D_{free} = 30 \; \mu m^2/min$ from maximal values obtained with GFP and corrected by the different Stokes radius. As above, we sought to minimize $L2(\theta) = (D_{exp}(s_{Vol}, Vol) - D_{sim}(\theta \mid s_{Vol}, Vol))^2$, where $D_{exp} (S_{Vol}, Vol)$

and $D_{sim}(\theta \mid s_{Vol}, Vol)$ are the experimental and simulated diffusion coefficients at a given growth rate and volume in G1 cells.

## Fitting procedures

For fitting Ydj1 diffusion and budding volume as a function of growth rate, we combined the respective square differences to seek $\mathrm{argmin}_\theta L1(\theta) + L2(\theta)$. To find the optimal value of $\theta$, we used a variant of the Levenberg–Marquardt algorithm with a second-order correction (Transtrum and Sethna, 2012). As all the simulations were performed at steady state, to obtain the Jacobian of the steady state with respect to the parameters, we applied the implicit function theorem following the approach described in https://arxiv.org/pdf/1602.02355.pdf. Numerical integration of the differential equations (to obtain an initial value for the steady state root finding) was performed using the *SciPy routine odeint*, which automatically switches between stiff and non-stiff solvers. After identifying the minima, we performed a Markov Chain Monte Carlo (Goodman & Weare, 2010) exploration of the parameter space using *emcee* (Foreman-Mackey et al, 2013). After discarding the initial 3,000 draws of the MC chain as burn-in, which were too close to the minima, we collected 1,500 sets with good match to data and repeated all simulations using this ensemble of parameters to estimate the uncertainty in parameters (Fig 3B) and simulations of variables used in the initial exhaustive fitting process (Fig 3F and G). Simulations of other variables that were experimentally tested (Figs 3H, 4B, and 5B) were performed with the parameter set 3114 (Table 1), which gave the minimal *P* value in the fitting process and using a fourfold range of values for the Ydj1 binding and release constant rates (*kb* and *kr*, respectively). CHX effects were simulated by reducing *ksProt*, *ksCln3*, and *ksYdj1* 3, 4.5, 6, 9, or 12-fold to test different conditions around the experimental reduction of 5.9-fold (Fig S7). Stress effects were simulated on steady states by transferring different fractions (20-80%) of the folded protein ($Prot_F$ and $Cln3_F$) to the unfolded population ($Prot_U$ and $Cln3_U$, respectively). To simulate transient effects in Cln3F by stress, experimental data of Ydj1-GFP mobility were used to fit the model time variable at different percentages of protein unfolding (Fig 5C and D), resulting in 60% for salt stress and 80% for heat stress. We took into account that heat, but not salt, stress increases Ydj1 mRNA levels by 2.9-fold (Gasch et al, 2000). Then, the resulting fitted parameters were applied to the model to simulate the evolution of Cln3F after stress, assuming that $Prot_U$ and $Cln3_U$ compete or not for $Ydj1_A$ (Fig 5G and H). The model was deposited in the BioModels (Chelliah et al, 2015) database as MODEL1808310001 in SBML format and a COPASI (Hoops et al, 2006) file to reproduce simulations with THE parameter set 3114. Code of the estimation of minima in IPython Notebooks is available upon request.

## Parameter distributions

As amply described in the literature, parameters in systems biology models are disheveled (Gutenkunst et al, 2007) or structurally unidentifiable (Szederkényi et al, 2011). We observed this behavior as well (Fig 3B), with parameters consistent with the experimental data spanning several orders of magnitude.

## Statistical analysis

We routinely show the standard error of the mean (N < 10) or confidence limits at $\alpha = 0.05$ (N > 10) to allow direct evaluation of variability and differences between mean values in plots. Sample size is always indicated in the figure legend and, when appropriate, *t* test *P* values are shown in the text. For model predictions, the mean values and standard deviations are plotted. All experiments were performed at least twice with fully independent cell samples.

## Miscellaneous

In vitro luciferase refolding assays have been described (Summers et al, 2009). Translational efficiency of refolding extracts was measured by incubation with 0.1 mM puromycin. Immunoblot analysis with $\alpha$puromycin (clone 12D10; Sigma-Aldrich), $\alpha$HA (clone 12CA5; Roche), and $\alpha$Dpm1 (clone 5C5; Molecular Probes) was as described (Georgieva et al, 2015). Ydj1 binding assays to GST fusions of Cln3, luciferase, and P6, a selected Ydj1-target peptide (Kota et al, 2009) used as reference, were performed with purified proteins as described (Lee et al, 2002). Protein synthesis rates in live cells were determined by $S^{35}$-methionine incorporation (Gallego et al, 1997).

# Supplementary Information

# Acknowledgements

We thank A Cornadó and E Rebollo for technical assistance, T Zimmermann for technical advice in FCS experiments, and B Futcher and J Skotheim for providing strains. We also thank C Rose for editing the manuscript and F Antequera, Y Barral, C Gallego, JC Igual, S Oliferenko, and F Posas for helpful comments. This work was funded by the Spanish Ministry of Science, Consolider-Ingenio 2010, and the European Union (FEDER) to M Aldea. DF Moreno received an FI fellowship from *Generalitat de Catalunya*.

## Author Contributions

DF Moreno: conceptualization, supervision, funding acquisition, investigation, and writing—original draft, review, and editing.
E Parisi: investigation.
G Yahya: investigation.
F Vaggi: investigation.
A Csikász-Nagy: conceptualization, supervision, investigation, and writing—original draft, review, and editing.
M Aldea: conceptualization, supervision, funding acquisition, investigation, and writing—original draft, review, and editing.

## Conflict of Interest Statement

The authors declare that they have no conflict of interest.

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
