## [Reviewer comments · Life Science Alliance]

Life Science Alliance

Competition in the chaperone-client network subordinates cell-cycle entry to growth and stress

David Moreno, Eva Parisi, Galal Yahya, Federico Vaggi, Attila Csikász-Nagy, and Marti Aldea

DOI: <https://doi.org/10.26508/lsa.201800277>

Corresponding author(s): Marti Aldea, Molecular Biology Institute of Barcelona IBMB-CSIC and Attila Csikász-Nagy, King's College London

Review Timeline:

Submission Date:	2018-12-12
Editorial Decision:	2019-01-18
Revision Received:	2019-04-01
Editorial Decision:	2019-04-08
Revision Received:	2019-04-08
Accepted:	2019-04-08

Scientific Editor: Andrea Leibfried

Transaction Report:

January 18, 2019

Re: Life Science Alliance manuscript #LSA-2018-00277-T

Martí Aldea
Institut de Biologia Molecular de Barcelona
Cell Biology
Baldri Reixac 15, 3 floor
Barcelona, Catalonia 08028
Spain

Dear Dr. Aldea,

Thank you for submitting your manuscript entitled "The chaperone-client network subordinates cell-cycle entry to growth and stress" to Life Science Alliance. The manuscript was assessed by expert reviewers, whose comments are appended to this letter.

As you will see, both reviewers appreciate your analyses and they support publication of a further revised version here. We would thus like to invite you to submit a revised version of your manuscript to us. The reviewers provide constructive input, and following their suggestions seems straightforward and aims at strengthening the current dataset and at making sure that the modeling part is correct and useful. We would thus expect that you address all concerns raised. Please also address Reviewer 2's point 3 and discuss in the manuscript whether previously proposed models like Whi5 dilution are or are not consistent with the proposed chaperone model.

Thank you for this interesting contribution to Life Science Alliance. We are looking forward to receiving your revised manuscript.

Sincerely,

- A letter addressing the reviewers' comments point by point.
- An editable version of the final text (.DOC or .DOCX) is needed for copyediting (no PDFs).
- High-resolution figure, supplementary figure and video files uploaded as individual files: See our detailed guidelines for preparing your production-ready images, <http://life-science-alliance.org/authorguide>
- Summary blurb (enter in submission system): A short text summarizing in a single sentence the study (max. 200 characters including spaces). This text is used in conjunction with the titles of papers, hence should be informative and complementary to the title and running title. It should describe the context and significance of the findings for a general readership; it should be written in the present tense and refer to the work in the third person. Author names should not be mentioned.

B. MANUSCRIPT ORGANIZATION AND FORMATTING:

Full guidelines are available on our Instructions for Authors page, <http://life-science-alliance.org/authorguide>

Reviewer #1 (Comments to the Authors (Required)):

Synopsis:

Faster growing yeast maintain cell size homeostasis at larger mass/volumes. The authors combine single cell microscopy techniques of fluorescent protein fusions (e.g. localization, FLIP, FCS) in mutant strains to provide evidence that competition of Cln3 for chaperone proteins can explain the connection between cell size threshold and growth rate, i.e. more unfolded proteins at faster growth rates titrate away chaperones from Cln3, which results in less nuclear Cln3 and, requires cells grow longer in G1 to reach the Start threshold. The beauty of the model is its simplicity and generality. The danger, however, is that chaperones have indirect effects on regulators other than Cln3 and, thus, the specific model could fall apart. The authors provide a lot of excellent data, they rule out trivial, indirect effects with good controls, and the results are compelling, interesting, and significant. No new data needed, except for *whi5*, *stb1*, *cln3* combinations in Fig. 2B (see my major comment).

Critiques:

** The data presentation is heavy. Better grouping, explanation, and justification in the captions (and other places in the manuscript) would help with readability; see comments below.

** The role of and justification for mathematical modeling was unclear to me; see comments below. Qualitative analysis of the experimental data could have established the same model shown in Figure 6.

Major comments:

** Figure 2B shows that chaperone overexpression affects size threshold in *cln3*, but not *whi5 stb1 cln3*. Cln3 is not the only regulator subordinate to chaperone abundance. It would be important to test other regulators (e.g. *Whi5*) and the magnitude of their effect relative to Cln3. Can the authors establish that Cln3 is the main regulator subordinate to chaperones?

** Where does growth rate (s_{vol}) enter into the mathematical model? This is an important detail because the data (i.e. growth rate, budding volume, effective diffusion coefficient of *Ydj1*) are used to constrain and infer model parameters. I am guessing that $s_{vol} = s_{protU}$, and that s_{cln3} , s_{ydj1} are fixed (but inferred) parameters. If true, what's the evidence and justification that total protein scales with growth rate, but *Ydj1* and Cln3 do not?

** Why are the authors using steady-state ratio of F_n , F_c to estimate K_e , K_i in Fig. 1F and EV2? They have a simple, dynamic mathematical model that should be fit to the *full* dynamic trace. The time-varying signal should better constrain the model and inferred parameters, right? I was confused by their use of steady-state and two different nuclear FLIP experiments to find K_e , K_i . Please elaborate.

Minor comments:

** A strength of the author's data is showing the full distribution of single cell data. More people should be following their example. Thus, why not show all data points in Figure 1D (similar to 1C) and Fig. 2A (similar to 2B)? Please clarify.

** Unclear to me how AZC affects Cln3 interactions with chaperones and, thus, hard to interpret data presented in Figures 1 and lines 103-108. Please elaborate and justify AZC experiment in Figure 1 caption and main text.

** Typo (auxin?) and/or missing experimental details in caption of 1H. Is 2NLS-GFP construct fused to "estradiol binding domain" and, thus, the authors mean "estradiol" rather than "auxin"? Please elaborate and justify experiment in Figure 1H caption, main text, and methods.

** Typo in Line 121 ("Fig. 2B represents budding size of newborn daughters"), or missing details in caption? The caption indicates that Fig. 2B is similar to Fig. 2A, except done for mutant backgrounds ... but C+H+S in Fig. 2A has different mean from wt C+H+S in Fig. 2B. Something is incorrect.

** Fig. 3A does not accurately reflect the elementary reactions of Eqs. (1-8). For example, YP and YC complex are not co-degraded (as shown in Fig. 3A). Rather each monomer unit is independently degraded and the partner is released back; thus, there should be two separate degradation arrows with partner released both for YP and YC complexes. More generally, the authors should better justify some of their choices and/or demonstrate that their choices don't affect model outcome. For example, the authors assume that unfolded protein and Cln3 irreversibly bind chaperone until either degradation and/or folding+release (i.e. no reverse, unbinding reaction). This seems a strong assumption that will affect the competition by putting it into the stoichiometric-binding regime. Same critique applies for independent degradation of subunits versus co-degradation.

** Line 361: How exactly was cell autofluorescence measured so that it could be subtracted from fluorescent strains? Please specify.

** Line 393-394: Unclear what is being corrected for with fluorescence data in non-bleached cell. Photobleaching due to fluorescent measurement? Please specify.

** Fig. 4A and 5A indicate that CHX or stress do not affect Ydj1 levels via synthesis or degradation? Typo? If not a typo, then please justify.

Reviewer #2 (Comments to the Authors (Required)):

Moreno et al. study an important property of life conserved from bacteria to humans, i.e. how cells regulate their size relative to their rate of growth (accumulation of mass), enabling cell size homeostasis. The authors add significant experimental evidence and mathematical modelling in favour of their previous hypothesis (Verges et al. Mol Cell 2007) that yeast cell size is controlled in late G1, at least in part, by Ydj1 chaperone-dependent release of Cln3 from the ER and subsequent nuclear accumulation. They propose a simple model whereby Cln3 competes with many cellular proteins for binding to limiting amounts of chaperones/co-chaperones needed for proper protein folding: when cells grow slowly and produce less proteins, enough Ydj1 is available for fast Cln3 folding and nuclear accumulation, causing cell cycle START at a small size; when cells grow fast in rich medium, Ydj1 and other chaperones are busy with many other client proteins, leading to delayed Cln3 nuclear accumulation and START at a larger size.

Cln3 is a low abundance and highly unstable G1 cyclin that is very difficult to observe in single yeast cells. To circumvent this problem, other authors had used a hypo-active but hyper-stable version of Cln3 (Cln3-11A) fused to a fluorescent protein, and disregarded the Cln3 ER retention model because they only saw Cln3-11A in the nucleus, never in the ER (Liu et al, 2015; Schmoller et al,

2015). Moreno et al. start by refuting this argument by showing that Ydj1 is important for nuclear accumulation even of Cln3-11A, but not of a reporter NLS-GFP construct (Fig. 1A-C). They also showed using FLIP that the nuclear import rate of Cdc28 in G1 is highly dependent on both Cln3 and Ydj1, whereas that of a reporter NLS-GFP construct is not. This is an important confirmation that Ydj1 plays a key and specific role for Cln3 and Cdc28 nuclear accumulation in G1.

To test the hypothesis that chaperones might be limiting for Cln3 nuclear accumulation, Moreno et al. measured cell size at budding (a marker of START) in strains containing one additional copy of genes coding for various sets of chaperone/co-chaperone (Ssa1+Ydj1, Hsc82+Cdc37, Cdc48+Ufd1+Npl4). Moderate overexpression of each set was sufficient to reduce the size at budding by 7-8%, whereas a combination of all 3 plasmids reduced this size by 17-35%. Crucially this effect was fully dependent on Cln3, Whi5 and Stb1, the main downstream effectors of START, and not on the upstream Whi7 regulator (Fig.2). Importantly the overall level and phosphorylation status of Cln3 were unchanged in cells containing the 3 additional CEN-chaperone plasmids (Fig EV3).

Next the authors develop a mathematical model where Cln3 competes with general protein synthesis for binding to Ydj1 to regulate the size at START. For this they use FCS to quantify the fraction of mobile Ydj1 as a measure of available (unbound) chaperone, and found indeed that Ydj1 availability decreases when cells are treated with a proteotoxic drug (AZC). Moreover Ydj1 mobility (availability) showed an inverse correlation with growth rate in individual G1 cells, indicating that the faster the growth, the more chaperones are busy folding other proteins (Fig.3). Consequently nuclear accumulation of Cln3-11A was also shown to be inversely proportional to growth rate, fitting the mathematical model (Fig.3H).

Conversely, reducing the workload of chaperones by reducing the protein synthesis rate using low doses of cycloheximide led to a faster Cln3 accumulation in the nucleus, which was Ydj1-dependent (Fig.4). Altogether the data point to the new concept that chaperone availability transmits the growth and protein synthesis rate information to modulate Cln3 nuclear accumulation and cell size at START of the cell cycle.

Finally, the authors show that Ydj1 availability and Cln3 nuclear accumulation decrease in several conditions of stress (heat, salt, ER), with a dynamics well reproduced by their mathematical model of chaperone titration (Fig.5). This explains the long-standing observations that various stresses inhibit transiently CLN1,2 transcription and budding.

The discussion paragraph puts the various concepts and data into perspective and offers a thoughtful explanation for the choice of free chaperone availability as a primary determinant for cell cycle Start, which would be to make sure that enough protein folding capacity is available in cells before the major biosynthetic wave that takes place soon after, during S phase.

Altogether this is a very interesting paper, clearly written and containing a number of clever, highly sophisticated and well-performed experiments. It proposes a new concept for a central question in cell biology, supported by solid evidence. Below are a few points that may deserve consideration, however, before publication.

Major points:

1. A prediction of the limiting chaperone model is that YDJ1 should be haplo-insufficient for size control, i.e. that a $ydj1\Delta/YDJ1$ heterozygous diploid (or $ydj1\Delta/YDJ1$ $ssa1\Delta/SSA1$) should have a larger size at budding than a wild-type diploid. This test is easy and should be included in the

revised ms. Also it would be nice to see if size reduction depends on the simultaneous increased gene dosage of chaperone and co-chaperone, i.e. if it still occurs when only Ydj1 gene dosage is increased.

2. It could be surprising that the levels of chaperones do not increase in parallel with the growth rate. Is there an explanation for this lack of coupling? Why don't cells contain higher concentrations of chaperones? What could be the selective pressure against it? Is there evidence that cells having 2-3 times more chaperones lose fitness or robustness?

3. Other mechanisms have been proposed to couple cell size at budding to cell growth, such as Whi5 dilution, decreased Cln3 threshold for Whi5 phosphorylation due to phosphatase inhibition by Rim15, or others that work independently of the main G1/S regulators Cln3, Whi5 and Bck1. It would help the reader to discuss these alternative concepts and see whether or not they fit with or complement the proposed chaperone titration model. A diagram taking into account these other possibilities might reflect better the reality.

Minor points:

1. Please indicate how Ydj1 and Ssa1 were overexpressed in Fig1D.
2. Why is the nuclear import rate of NLS-GFP not shown for ydj1 Δ cells in Fig.1H?
3. Why does C+H+S cause a size reduction of 17% in Fig.2A and of ~35% in Fig. 2B?
4. correct Line 284: If this modulation ...
5. correct Line 419: Ydj1 is a stable...
6. The depth and impact of the paper might deserve a catchier title.

Reviewer #1 (Comments to the Authors (Required)):

Synopsis:

Faster growing yeast maintain cell size homeostasis at larger mass/volumes. The authors combine single cell microscopy techniques of fluorescent protein fusions (e.g. localization, FLIP, FCS) in mutant strains to provide evidence that competition of Cln3 for chaperone proteins can explain the connection between cell size threshold and growth rate, i.e. more unfolded proteins at faster growth rates titrate away chaperones from Cln3, which results in less nuclear Cln3 and, requires cells grow longer in G1 to reach the Start threshold. The beauty of the model is its simplicity and generality. The danger, however, is that chaperones have indirect effects on regulators other than Cln3 and, thus, the specific model could fall apart. The authors provide a lot of excellent data, they rule out trivial, indirect effects with good controls, and the results are compelling, interesting, and significant. No new data needed, except for whi5, stb1, cln3 combinations in Fig. 2B (see my major comment).

Critiques:

*** The data presentation is heavy. Better grouping, explanation, and justification in the captions (and other places in the manuscript) would help with readability; see comments below.*

*** The role of and justification for mathematical modeling was unclear to me; see comments below. Qualitative analysis of the experimental data could have established the same model shown in Figure 6.*

Major comments:

*** Figure 2B shows that chaperone overexpression affects size threshold in cln3, but not whi5 stb1 cln3. Cln3 is not the only regulator subordinate to chaperone abundance. It would be important to test other regulators (e.g. Whi5) and the magnitude of their effect relative to Cln3. Can the authors establish that Cln3 is the main regulator subordinate to chaperones?*

We have analyzed the effects of increased chaperone-gene copy number in whi5 cells, and the data are plotted in new Fig 2B. Whi5-deficient cells exhibited a small decrease in budding size in the presence of plasmids expressing the three chaperone sets, but less pronounced than that shown by wt cells. Since whi5 cells still require Cln3 to attain their small size (Jorgensen et al., 2002), the observed residual decrease could be due to Cln3-mediated effects. We have added these comments to the revised version of the manuscript (L130-133).

*** Where does growth rate (s_{vol}) enter into the mathematical model? This is an important detail because the data (i.e. growth rate, budding volume, effective diffusion coefficient of Ydj1) are used to constrain and infer model parameters. I am guessing that $s_{vol} = s_{prot}U$, and that s_{cln3} , s_{ydj1} are fixed (but inferred) parameters. If true, what's the evidence and justification that total protein scales with growth rate, but Ydj1 and Cln3 do not?*

The description of the model parameters has been modified to clarify this point. (L434-440). We assume that ProtU synthesis rate is proportional to the cell volume (Vol), as well as to the cell-specific volume growth rate (s_{Vol}), which can be expressed as $s_{Vol} = s_{Prot} \cdot \gamma$, where s_{Prot} is the cell-specific protein synthesis rate and γ is a scale conversion factor. Because of lack of data and to

avoid over-fitting γ was fixed at 1. Regarding s_{Cln3} and s_{Ydj1} , we show that the overall concentration of mCitrine-Cln311A and Ydj1-GFP proteins expressed under endogenous transcriptional and translational regulatory sequences does not change with growth rate in G1 at the single cell level (Fig S6G,H). This is in agreement with the finding that CLN3 and YDJ1 mRNA levels relative to total mRNA are quite constant at different growth rates in chemostats limited by different nutrients (Slavov et al, 2011). Thus, to keep the model as simple as possible, we opted for keeping Cln3 and Ydj1 synthesis rates unaltered during the calculations to obtain every steady state at different overall protein synthesis rates without altering the levels of Cln3 and Ydj1. Indeed, although it is not the case in live cells, making Ydj1 concentration proportional to growth rate in the model abrogates the dependence of Cln3F. In contrast, when Cln3 concentration was made proportional to the overall protein synthesis rate in the model, Cln3F and Cln3U levels displayed the same relative dependency on growth rate shown in Supplementary Fig S6E.

**** Why are the authors using steady-state ratio of F_n , F_c to estimate K_e , K_i in Fig. 1F and EV2? They have a simple, dynamic mathematical model that should be fit to the *full* dynamic trace. The time-varying signal should better constrain the model and inferred parameters, right? I was confused by their use of steady-state and two different nuclear FLIP experiments to find K_e , K_i . Please elaborate.**

The very initial drop in fluorescence is much pronounced in the nucleus than in the cytoplasm, indicating that a fraction of the nuclear molecules is not in “steady-state” regarding import/export kinetics. This could be due, for instance, to binding to chromatin. Taking this into account, we have followed the suggestion of the reviewer and fitted observed data after dropping the initial 6 seconds after bleaching. For each cell the bleaching constant was obtained by fitting Htb2-mCherry data, and the import rate (relative to export rate) by fitting Cdc28-GFP data from wild type and *cln3* cells. The figure below contains representative fits (panels A and B) and shows that obtained import rates were comparable to those from steady-state data (panels C and D).

Minor comments:

**** A strength of the author's data is showing the full distribution of single cell data. More people should be following their example. Thus, why not show all data points in Figure 1D (similar to 1C) and Fig. 2A (similar to 2B)? Please clarify.**

We thank the reviewer for noticing the error in Fig 1D (white boxes where on top of single-cell data, which has now been corrected). Regarding 2A we would prefer to keep the bar plot as it is because the differences observed in asynchronous cells were not as large as with newborn cells shown in Fig 2B, where we display the whole set of single-cell data.

**** Unclear to me how AZC affects Cln3 interactions with chaperones and, thus, hard to interpret data presented in Figures 1 and lines 103-108. Please elaborate and justify AZC experiment in Figure 1 caption and main text.**

Azetidine 2-carboxylic acid (AZC) is a proline analog that interferes with proper protein folding (Trotter et al, 2001) and causes large aggregates of misfolded proteins that sequester Ssa1 and other chaperones (Escusa-Toret et al, 2013). This is the reason why we used this drug to decrease the levels of available Ssa1 and (likely) Ydj1. We have modified the text to clarify this point (L106-107).

**** Typo (auxin?) and/or missing experimental details in caption of 1H. Is 2NLS-GFP construct fused to "estradiol binding domain" and, thus, the authors mean "estradiol" rather than "auxin"? Please elaborate and justify experiment in Figure 1H caption, main text, and methods.**

As the reviewer pinpointed there was a mistake in legend for Fig 1H. To test indirect effects through the constitutive nuclear import machinery we had analyzed a 2NLS-GFP construct fused to the estradiol-binding domain immediately after the addition of estradiol to allow nuclear import. We have corrected Fig 1H legend and added this comment to the text to clarify the objective of the experiment (L107-111).

**** Typo in Line 121 ("Fig. 2B represents budding size of newborn daughters"), or missing details in caption? The caption indicates that Fig. 2B is similar to Fig. 2A, except done for mutant backgrounds ... but C+H+S in Fig. 2A has different mean from wt C+H+S in Fig. 2B. Something is incorrect.**

We apologize for the missing information in legend for Fig 2. As stated in the text, the first analyses in Fig 2A were done with asynchronous cells for reasons of simplicity, but those shown in Fig 2B and C were done with daughter (or newly-born) cells isolated by differential centrifugation. We have added this information to the legend of Fig 2.

**** Fig. 3A does not accurately reflect the elementary reactions of Eqs. (1-8). For example, YP and YC complex are not co-degraded (as shown in Fig. 3A). Rather each monomer unit is independently degraded and the partner is released back; thus, there should be two separate degradation arrows with partner released both for YP and YC complexes. More generally, the authors should better justify some of their choices and/or demonstrate that their choices don't affect model outcome. For example, the authors assume that unfolded protein and Cln3 irreversibly bind chaperone until either degradation and/or folding+release (i.e. no reverse, unbinding reaction). This seems a strong assumption that will affect the competition by putting it into the stoichiometric-**

binding regime. Same critique applies for independent degradation of subunits versus co-degradation.

We wanted to keep the number of parameters to fit at the minimal. Following this goal, we did not assume co-degradation of both molecules from protein complexes. Rather we assumed that their degradation rate is the same in both free and bound forms. Now we expanded the Model parameters and simulations section with these details (L450-452), and Fig 3A has been modified to clearly indicate this assumption. We assumed unidirectional binding of chaperones to target proteins also for simplicity reasons. In any event, we tested the effects of a slow unbinding reaction (1% or 10% of binding) from the complex and the results shown in Fig S6D-F were not affected qualitatively.

*** Line 361: How exactly was cell autofluorescence measured so that it could be subtracted from fluorescent strains? Please specify.*

Autofluorescence levels were obtained in the same experimental scenario from naive cells (with same genotype but not expressing the indicated GFP or mCherry fusion protein). This comment has been added to the text (L383).

*** Line 393-394: Unclear what is being corrected for with fluorescence data in non-bleached cell. Photobleaching due to fluorescent measurement? Please specify.*

As the reviewer guessed, the small loss of fluorescence caused by image acquisition was measured in control cells in the same field that were not purposely bleached with the laser at full power. This comment has been added to the text (L398-400).

*** Fig. 4A and 5A indicate that CHX or stress do not affect Ydj1 levels via synthesis or degradation? Typo? If not a typo, then please justify.*

As for comment 2, and also because Ydj1 levels do not change during the short course of these experiments, we opted for maintaining Ydj1 levels constant in the model.

Reviewer #2 (Comments to the Authors (Required)):

Moreno et al. study an important property of life conserved from bacteria to humans, i.e. how cells regulate their size relative to their rate of growth (accumulation of mass), enabling cell size homeostasis. The authors add significant experimental evidence and mathematical modelling in favour of their previous hypothesis (Verges et al. Mol Cell 2007) that yeast cell size is controlled in late G1, at least in part, by Ydj1 chaperone-dependent release of Cln3 from the ER and subsequent nuclear accumulation. They propose a simple model whereby Cln3 competes with many cellular proteins for binding to limiting amounts of chaperones/co-chaperones needed for proper protein folding: when cells grow slowly and produce less proteins, enough Ydj1 is available for fast Cln3 folding and nuclear accumulation, causing cell cycle START at a small size; when cells grow fast in rich medium, Ydj1 and other chaperones are busy with many other client proteins, leading to delayed Cln3 nuclear accumulation and START at a larger size.

Cln3 is a low abundance and highly unstable G1 cyclin that is very difficult to observe in single yeast cells. To circumvent this problem, other authors had used a hypo-active but hyper-stable version of Cln3 (Cln3-11A) fused to a fluorescent protein, and disregarded the Cln3 ER retention model because they only saw Cln3-11A in the nucleus, never in the ER (Liu et al, 2015; Schmoller et al, 2015). Moreno et al. start by refuting this argument by showing that Ydj1 is important for nuclear accumulation even of Cln3-11A, but not of a reporter NLS-GFP construct (Fig. 1A-C). They also showed using FLIP that the nuclear import rate of Cdc28 in G1 is highly dependent on both Cln3 and Ydj1, whereas that of a reporter NLS-GFP construct is not. This is an important confirmation that Ydj1 plays a key and specific role for Cln3 and Cdc28 nuclear accumulation in G1.

To test the hypothesis that chaperones might be limiting for Cln3 nuclear accumulation, Moreno et al. measured cell size at budding (a marker of START) in strains containing one additional copy of genes coding for various sets of chaperone/co-chaperone (Ssa1+Ydj1, Hsc82+Cdc37, Cdc48+Ufd1+Npl4). Moderate overexpression of each set was sufficient to reduce the size at budding by 7-8%, whereas a combination of all 3 plasmids reduced this size by 17-35%. Crucially this effect was fully dependent on Cln3, Whi5 and Stb1, the main downstream effectors of START, and not on the upstream Whi7 regulator (Fig.2). Importantly the overall level and phosphorylation status of Cln3 were unchanged in cells containing the 3 additional CEN-chaperone plasmids (Fig EV3).

Next the authors develop a mathematical model where Cln3 competes with general protein synthesis for binding to Ydj1 to regulate the size at START. For this they use FCS to quantify the fraction of mobile Ydj1 as a measure of available (unbound) chaperone, and found indeed that Ydj1 availability decreases when cells are treated with a proteotoxic drug (AZC). Moreover Ydj1 mobility (availability) showed an inverse correlation with growth rate in individual G1 cells, indicating that the faster the growth, the more chaperones are busy folding other proteins (Fig.3). Consequently nuclear accumulation of Cln3-11A was also shown to be inversely proportional to growth rate, fitting the mathematical model (Fig.3H).

Conversely, reducing the workload of chaperones by reducing the protein synthesis rate using low doses of cycloheximide led to a faster Cln3 accumulation in the nucleus, which was Ydj1-dependent (Fig.4). Altogether the data point to the new concept that

chaperone availability transmits the growth and protein synthesis rate information to modulate Cln3 nuclear accumulation and cell size at START of the cell cycle.

Finally, the authors show that Ydj1 availability and Cln3 nuclear accumulation decrease in several conditions of stress (heat, salt, ER), with a dynamics well reproduced by their mathematical model of chaperone titration (Fig.5). This explains the long-standing observations that various stresses inhibit transiently CLN1,2 transcription and budding.

The discussion paragraph puts the various concepts and data into perspective and offers a thoughtful explanation for the choice of free chaperone availability as a primary determinant for cell cycle Start, which would be to make sure that enough protein folding capacity is available in cells before the major biosynthetic wave that takes place soon after, during S phase.

Altogether this is a very interesting paper, clearly written and containing a number of clever, highly sophisticated and well-performed experiments. It proposes a new concept for a central question in cell biology, supported by solid evidence. Below are a few points that may deserve consideration, however, before publication.

Major points:

1. A prediction of the limiting chaperone model is that YDJ1 should be haplo-insufficient for size control, i.e. that a ydj1 Δ /YDJ1 heterozygous diploid (or ydj1 Δ /YDJ1 ssa1 Δ /SSA1) should have a larger size at budding than a wild-type diploid. This test is easy and should be included in the revised ms. Also it would be nice to see if size reduction depends on the simultaneous increased gene dosage of chaperone and co-chaperone, i.e. if it still occurs when only Ydj1 gene dosage is increased.

Following the interesting suggestion given by the reviewer, we have carefully analyzed Ydj1 protein levels by WB with a Ydj1-specific antibody in YDJ1/ydj1 heterozygous diploid cells. As shown in new Fig S3, a reduction to ca. 60% in Ydj1 protein levels caused a significant and reproducible increase in budding size, thus supporting the limiting role of chaperones in setting cell size during cell-cycle entry (L145-149). We thank the reviewer for the nice experiment suggested.

Regarding chaperone co-expression experiments, we had first analyzed the effects of Ssa1 and Ydj1, and observed that the budding volume of newborn cells with two copies of Ssa1 or Ssa1/Ydj1 was 5% and 11% smaller, respectively, compared to those with empty vector. We have added this comment to the manuscript (L118-120).

2. It could be surprising that the levels of chaperones do not increase in parallel with the growth rate. Is there an explanation for this lack of coupling? Why don't cells contain higher concentrations of chaperones? What could be the selective pressure against it? Is there evidence that cells having 2-3 times more chaperones lose fitness or robustness?

We show that the overall concentration of Ydj1-GFP expressed under endogenous transcriptional and translational regulatory sequences does not change with growth rate in G1 at the single cell level (new Fig S6G). This is in agreement with the finding that YDJ1 (and SSA1/2) mRNA levels relative to total mRNA are quite constant at different growth rates in chemostats limited by different nutrients (Slavlov et al., 2011). On the other hand, overexpression of YDJ1 and SSA1 chaperones from the GAL1p promoter, which only attains a moderate 50-60% increase in Ydj1 levels (Yahya et al., 2007), produces an

abnormally enlarged morphology in budded mother cells after two or three generations (our unpublished data), suggesting that other phases of the cell cycle might be sensitive to excess of Ssa1-Ydj1 chaperone activity, perhaps by spurious interference with complex formation of proteins involved in mitosis and/or cytokinesis.

3. Other mechanisms have been proposed to couple cell size at budding to cell growth, such as Whi5 dilution, decreased Cln3 threshold for Whi5 phosphorylation due to phosphatase inhibition by Rim15, or others that work independently of the main G1/S regulators Cln3, Whi5 and Bck1. It would help the reader to discuss these alternative concepts and see whether or not they fit with or complement the proposed chaperone titration model. A diagram taking into account these other possibilities might reflect better the reality.

We discuss in the manuscript the possible interactions of our model with Whi5 dilution in G1 by growth (L270-277 and L303-311) and Rim15-mediated dephosphorylation of Whi5 as a modulator of cell-cycle entry at low growth rates in poor-carbon sources (L290-291). As suggested by the reviewer, we have modified the final schematic to include the role of Whi5 and Rim15 and, at the same time, show more explicit models for growth and stress-mediated effects on Start (new Fig. 6A,B).

Minor points:

1. Please indicate how Ydj1 and Ssa1 were overexpressed in Fig1D.

Ydj1 and Ssa1 (oYdj1 oSsa1) were overexpressed from the dual GAL1-10p promoter in Fig1D. The figure legend has been modified to include this information.

2. Why is the nuclear import rate of NLS-GFP not shown for ydj1Δ cells in Fig.1H?

Fig1C shows that the ydj1 deletion does not affect mCitrine-Cln3-11A nuclear accumulation, and we felt that this control would be unnecessary in Fig1H.

3. Why does C+H+S cause a size reduction of 17% in Fig.2A and of ~35% in Fig. 2B?

Fig 2A shows asynchronous cells, while 2B shows newborn cells. The figure legend has been corrected to clarify this point. As mentioned above, chaperone overexpression produces an abnormally enlarged morphology in budded mother cells after two or three generations, which would increase the average budding volume of the asynchronous population.

4. correct Line 284: If this modulation ...

The mistake has been corrected (L303).

5. correct Line 419: Ydj1 is a stable...

The mistake has been corrected (L447).

6. The depth and impact of the paper might deserve a catchier title.

Thanks for the proposal. We have added the word “competition” to attract the more “liberal” audience... ;)

April 8, 2019

RE: Life Science Alliance Manuscript #LSA-2018-00277-TR

Dr. Marti Aldea
Molecular Biology Institute of Barcelona IBMB-CSIC
Baldiri Reixac 15, 3-A16
Barcelona, Catalonia 08028
Spain

Dear Dr. Aldea,

Thank you for submitting your revised manuscript entitled "Competition in the chaperone-client network subordinates cell-cycle entry to growth and stress". As you will see, the reviewers appreciate the introduced changes and we would thus be happy to publish your paper in Life Science Alliance pending final revisions necessary to meet our formatting guidelines:

- please upload all figures (including suppl figures) as individual files
- please fill in the electronic license to publish form

A. FINAL FILES:

B. MANUSCRIPT ORGANIZATION AND FORMATTING:

Sincerely,

Reviewer #1 (Comments to the Authors (Required)):

The authors have done an excellent job addressing all questions and critiques. This is a very good and thoughtful paper. I look forward to seeing it published soon.

Reviewer #2 (Comments to the Authors (Required)):

This reviewer feels that the authors have answered satisfactorily to most, if not all, points raised by the reviewers. The manuscript is even stronger now and should be published.

April 8, 2019

RE: Life Science Alliance Manuscript #LSA-2018-00277-TRR

Dr. Marti Aldea
Molecular Biology Institute of Barcelona IBMB-CSIC
Baldiri Reixac 15, 3-A16
Barcelona, Catalonia 08028
Spain

Dear Dr. Aldea,

Thank you for submitting your Research Article entitled "Competition in the chaperone-client network subordinates cell-cycle entry to growth and stress". It is a pleasure to let you know that your manuscript is now accepted for publication in Life Science Alliance. Congratulations on this interesting work.

DISTRIBUTION OF MATERIALS:

Again, congratulations on a very nice paper. I hope you found the review process to be constructive and are pleased with how the manuscript was handled editorially. We look forward to future exciting submissions from your lab.

Sincerely,
